# Activation of serotonin neurons promotes active persistence in a probabilistic foraging task

Eran Lottem [1], Dhruba Banerjee[2], Pietro Vertechi[1], Dario Sarra[1], Matthijs oude Lohuis [3] & Zachary F. Mainen[1]

The neuromodulator serotonin (5-HT) has been implicated in a variety of functions that involve patience or impulse control. Many of these effects are consistent with a long-standing theory that 5-HT promotes behavioral inhibition, a motivational bias favoring passive over active behaviors. To further test this idea, we studied the impact of 5-HT in a probabilistic foraging task, in which mice must learn the statistics of the environment and infer when to leave a depleted foraging site for the next. Critically, mice were required to actively nose-poke in order to exploit a given site. We show that optogenetic activation of 5-HT neurons in the dorsal raphe nucleus increases the willingness of mice to actively attempt to exploit a reward site before giving up. These results indicate that behavioral inhibition is not an adequate description of 5-HT function and suggest that a unified account must be based on a higher-order function.

[1] Champalimaud Research, Champalimaud Centre for the Unknown, 1400-038 Lisbon, Portugal. [2] School of Medicine, University of California, Irvine, CA 92697-3950, USA. [3] Swammerdam Institute for Life Sciences, Center for Neuroscience, Faculty of Science, University of Amsterdam, 1098XH Amsterdam, The Netherlands. Correspondence and requests for materials should be addressed to Z.F.M. (email: zmainen@neuro.fchampalimaud.org)

Serotonin (5-HT) is a central neuromodulator implicated in the regulation of many biological processes and is one of the most important targets of psychoactive drugs[1, 2]. As a unifying concept for 5-HT's manifold effects, Soubrié[3] put forward the hypothesis that a major function of 5-HT is to promote behavioral inhibition. Building on work showing that blockade of 5-HT transmission results in continued responses to stimuli that are no longer rewarding[4–7], he argued that 5-HT biases decisions in favor of passive over active responding.

More recently, the study of 5-HT and behavioral inhibition has concentrated chiefly on impulse control[8–10]. One of the most common tasks used to study impulsive behavior is the five-choice serial reaction time task (5-CSRTT)[11]. Although this task was not specifically designed to measure impulsivity, animals sometimes respond before stimulus presentation, a behavior indicative of impulsivity, and brain-wide 5-HT depletion increases impulsivity in this task[12].

Another line of experiments focuses on animals' ability to wait in order to obtain reward[13]. Neurons in the dorsal raphe nucleus (DRN; the major source of 5-HT to the forebrain) increase their firing rates during reward waiting[14]. Moreover, pharmacological inhibition of these neurons promotes premature leaving[14], whereas optogenetic activation of the same neurons promotes patience[15, 16].

All of these results are broadly consistent with the Soubrié theory of behavioral inhibition. By reducing the motivation to act, increased 5-HT levels would reduce the rate of premature responses in the 5-CSRTT and increase the time animals can wait to obtain delayed rewards. However, they are also consistent with a different theory in which 5-HT promotes not passivity or the ability to tolerate inaction, but the patience or persistence to carry out an action that is itself costly or unrewarding but helps lead to delayed benefit. Thus successful waiting, in this alternative account, consists not only in suppressing the urge to respond prematurely but also actively carrying out a behavioral alternative to responding. Indeed, children facing the "marshmallow task", in which they must keep from eating one tasty treat in order to gain a second one later on, often succeed not only by passively controlling their impulse to eat the available marshmallow but also by actively distracting themselves by performing alternative behaviors[17]. So far, in the tasks used to study 5-HT, waiting and passivity coincide by design and so these alternative explanations cannot be readily distinguished. Indeed, DRN neurons fire during some types of movements[18] and DRN 5-HT neurons show increased activity during reward consumption compared to travel[19], consistent with an involvement in active behaviors.

Here we sought to disambiguate whether 5-HT promotes waiting through behavioral inhibition or though persistence in the context of foraging behavior. In natural environments, resources are typically found in patches that become exhausted over time, facing animals with an exploitation/exploration dilemma: they must choose between working within in a given patch and giving up to travel to a different one. Foraging within a patch is itself subject to uncertainty—that is the income from a patch is itself irregular or probabilistic—then exploitation requires patience. But this kind of patience is active rather than passive: in order to obtain rewards animals must continue to seek for food even when food-gathering attempts are sometimes unsuccessful.

There are a number of studies of optimal foraging in rodents in their natural habitats[20, 21], and it has been proposed that certain operant behaviors are closely related to foraging behavior[22], but the use of foraging behaviors in a laboratory setting remains a relatively underexplored area (but see refs. [23, 24]). We therefore developed a novel probabilistic foraging task in which mice nose-poke (forage) at one of two ports and are rewarded for their pokes (foraging attempts) according to a random probability schedule that decays exponentially to zero. Owing to the probabilistic nature of the reward schedule, in each trial, a mouse will experience rewarded pokes interspersed with unrewarded pokes. We reasoned that if 5-HT promotes behavioral inhibition and passive behavior, then it should suppress poking, resulting in fewer pokes at each port visit. If, instead, 5-HT promotes persistence, it should promote the active behavior, nose-poking, which is required to obtain rewards.

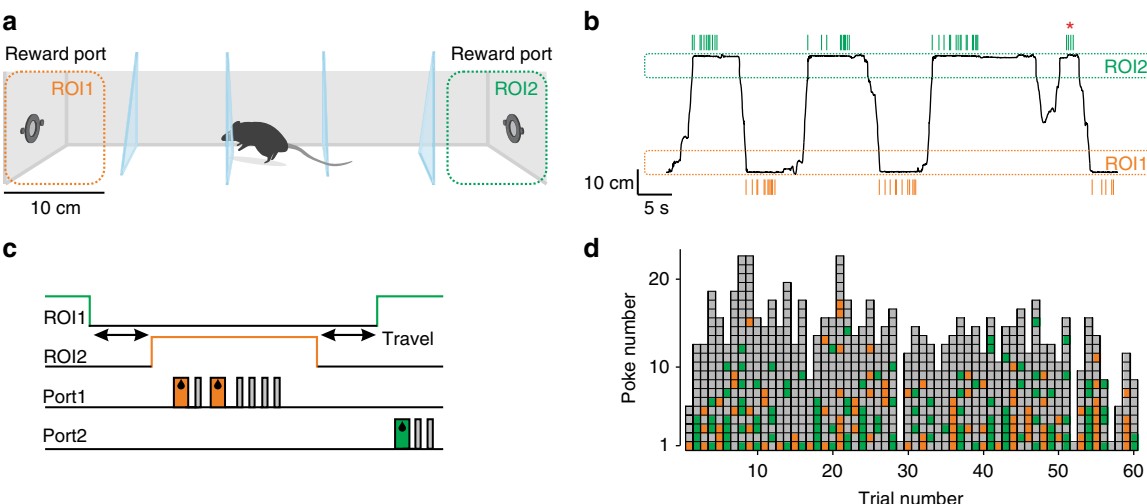

**Fig. 1** The probabilistic foraging task. **a** Schematic drawing of the foraging task apparatus. Mice shuttle back and forth between two reward sites, located at the opposite ends of an elongated box, to obtain water rewards. **b** Example snapshot of foraging behavior. The one-dimensional location of an example mouse along the long axis of the box is plotted as a function of time. The ROIs around each water port are marked as dashed rectangles, and green and orange ticks above and below the trajectory mark nose-pokes into the right and left ports, respectively. The red asterisk marks an error trial. **c** Task events during a single trial. Each trial starts with an exit from one of the ROIs. Following shuttling to the other end, the mice would nose-poke multiple times and receive reward on some of the attempts on a probabilistic basis, before switching back. Green and orange rectangles mark rewards, gray rectangles mark omissions. **d** Example session, this time showing only the sequence of outcomes during nose-poking. Each column represents a single trial. Green/orange squares represent rewarded nose-pokes, gray squares, omissions

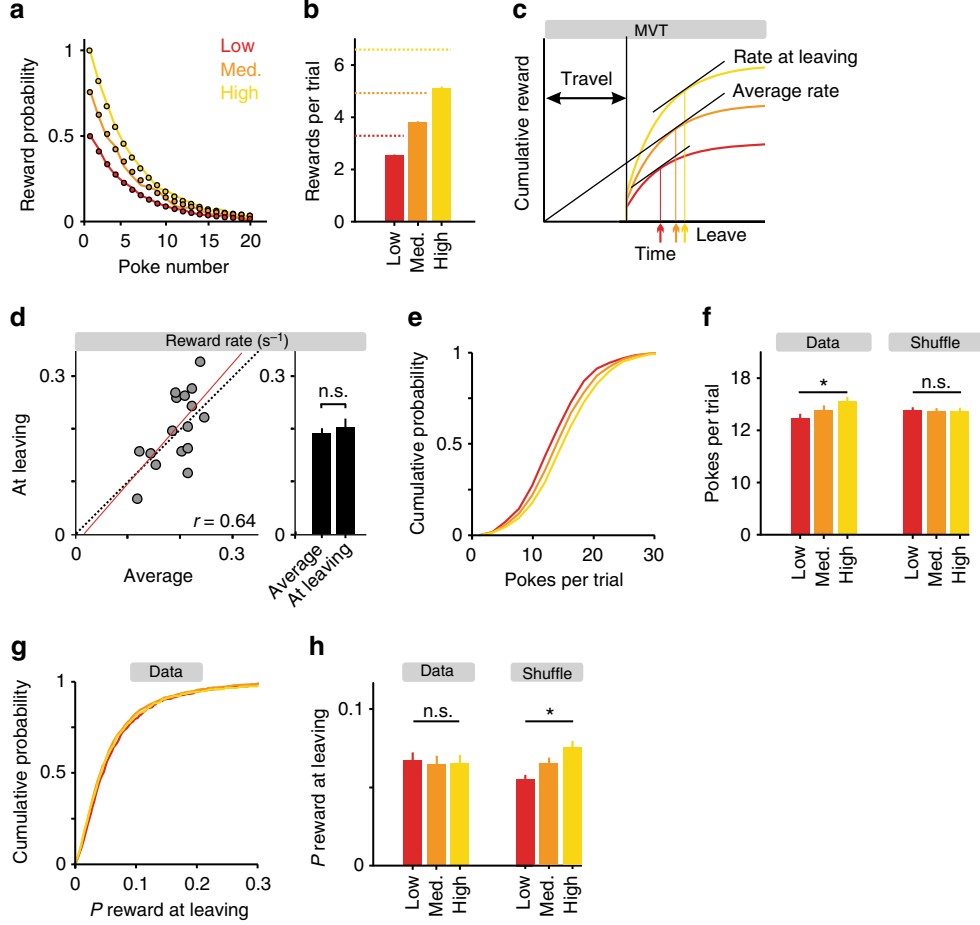

**Fig. 2** Reward statistics and task performance. **a** In each trial, reward probabilities were drawn from one of the three exponentially decreasing functions, all sharing the same time constant but with a different scaling factor, labeled high, medium, and low. Dots mark hypothetical values; solid lines are averages derived from data. **b** Bar plot showing the average number of rewards in each of the trial types ($n = 16$ mice). Dashed lines mark maximal values (assuming mice stay in place and poke indefinitely). **c** Schematic drawing of an optimal-agent's behavior during foraging. When plotting the average cumulative reward as a function of time from reward site exit, the average reward rate is the slope of the line that connects this curve at the time of leaving with the origin. Thus the slope of this line is maximal when it is tangent to the curve. Consequently, better or worse trials result in later or earlier leaving times, respectively (vertical lines and arrows). **d** Left: Scatter plot of reward rate at leaving vs. average reward rate. Each circle represents one mouse ($n = 16$). Dashed line is the unity diagonal and red line is a linear regression curve, with its correlation coefficient shown as well ($p < 0.001$). Right: Bar plot showing the average reward rate at leaving and the average reward rate. $p > 0.05$, Wilcoxon signed-rank test. **e** Cumulative distributions of the number of pokes per trial for the three trial types, averaged across mice ($n = 16$). **f** Bar plot showing the average number of pokes in each of the trial types. Bars on the left represent real data, and bars on the right represent shuffled data. Asterisk indicates significant effect ($p < 0.05$, ANOVA). **g** Cumulative distributions of the estimated reward probability after the last poke in a trial (i.e., at the time of switching) for the three trial types, averaged across mice ($n = 16$). **h** Bar plot showing the average reward probability after the last poke. Bars on the left represent real data, and bars on the right represent shuffled data. Asterisk indicates significant difference between trial types, $p < 0.05$, ANOVA

We found that optogenetic activation of DRN 5-HT neurons increases the number of active nose-pokes a mouse would carry out in an attempt to gain water before giving up. These results contradict the behavioral inhibition hypothesis and support the notion that 5-HT promotes waiting by enhancing persistence in the face of uncertainty and delay.

## Results

**Mouse behavior in a probabilistic foraging task.** To study the role of 5-HT in foraging mice, we developed a novel probabilistic foraging task. Water-restricted mice were placed in a rectangular chamber (50 cm long) containing one water port at each end, which acted as foraging sites (Fig. 1a). Body position was determined using video tracking. A foraging "trial" was considered as a visit to one port, defined as the period between entry and exit into

a region of interest (ROI) around each port (Fig. 1b, c). Correct trials were ones in which the mouse alternated between ports, whereas error trials were those in which the mouse left and then re-entered the same ROI. In correct trials, each nose-poke into the port was either rewarded or not according to a defined probability schedule (Fig. 1d). Reward probabilities were reset to their highest value at the start of each trial and declined exponentially with each poke. Error trials were unrewarded.

To incentivize goal-directed behavior, we introduced three trial types with different initial probability of reward (high-, medium-, and low-quality patches), so that mice would benefit by taking into account the actual rewards received in a given trial, as opposed to adopting a fixed, reward-independent strategy. Reward probability decayed exponentially at the same rate regardless of initial probability (Fig. 2a; Eq. 1 in Methods). The

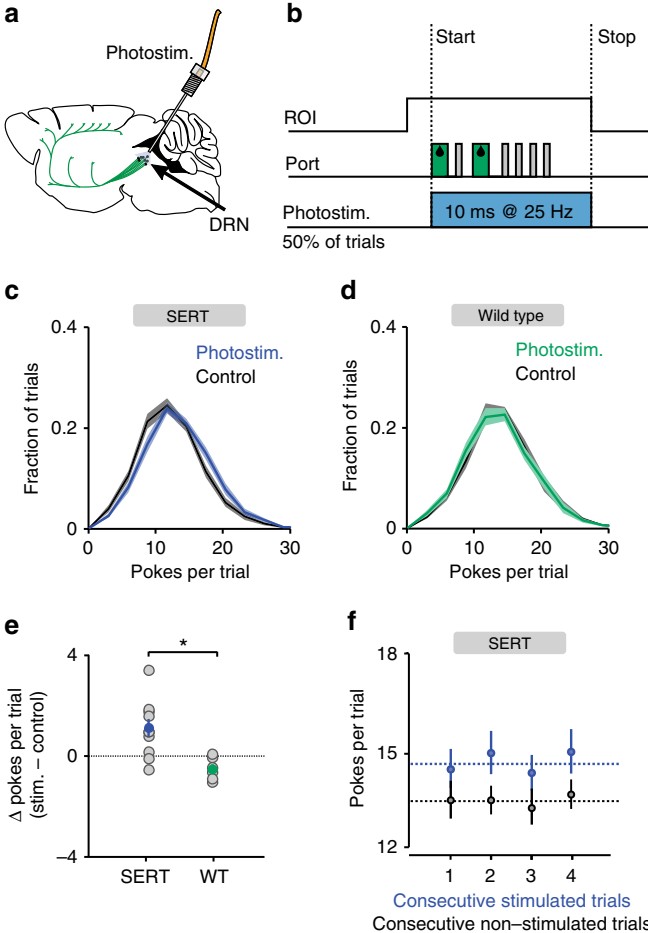

**Fig. 3** The effect of DRN 5-HT photostimulation on switching behavior. **a** Scheme of the locations of ChR2-YFP expression and optic fiber placement (adapted with permission from ref. [63]). **b** Schematic diagram of task events during a single trail, also showing the period of photostimulation. Photostimulation was triggered by the first poke in 50% of correct trials and ended either when the mouse left the ROI or if 10 s had elapsed since the last poke. **c** Distributions of the number of pokes per trial for photostimulated trials (blue) and control trials (black) averaged across the population of SERT-Cre mice ($n = 10$). **d** Distributions of the number of pokes per trial for photostimulated (green) and control (black) trials, across the population of wild-type mice ($n = 6$). **e** Difference between average number of pokes in photostimulated and control trials for SERT-Cre ($n = 10$) and wild-type ($n = 6$) mice. Averages across mice are shown in filled circles. *$p < 0.05$, Wilcoxon rank-sum test. **f** Plot showing the average number of pokes in photostimulated and control trials for SERT-Cre mice ($n = 10$) as a function of consecutive numbers of photostimulated (blue) or control (black) trials. There was no effect of sequence length on behavior, $p > 0.05$, linear regression analysis

three trial types were presented in a randomly interleaved manner and were not cued. As expected, the average number of rewards gained was highest in high-quality patches and lowest in low-quality patches (Fig. 2b). However, the number of rewards mice obtained was lower than the total available. This could be expected due to a trade-off between continuing to try to exploit the current port and leaving to try the other, at the cost of travel between ports ($3.32 \pm 1.70$ s (mean ± SD)). This tradeoff is formalized within optimal foraging theory by the marginal value theorem (MVT)[25], which states that optimal foragers should leave a depleting resource whenever the instantaneous reward rate

within a patch drops below the long-term average reward rate calculated taking into account travel times (Fig. 2c).

We tested the MVT in our data set and found that the reward rate at the time of leaving was indeed nearly identical to the average reward rate ($0.19 \pm 0.010$ rewards s$^{-1}$ on average and $0.20 \pm 0.017$ rewards s$^{-1}$ at leaving (mean ± SEM); $p = 0.54$, Wilcoxon signed-rank test, $n = 16$ mice; Fig. 2d). Two additional predictions of the MVT are that[26]: (1) the mice should make more pokes in better than average trials, and less in worse than average trials; and (2) reward probability at the time of leaving should be the same regardless of the initial probability at the start of the trial. We tested these predictions in our data set by comparing the number of pokes made (Fig. 2e, f) and the expected reward probabilities at the time of leaving (Fig. 2g, h) for the three trial types. We found that, indeed, the number of pokes increased with increasing initial probability ($F_{(2,45)} = 3.49$, $p = 0.04$, one-way analysis of variance (ANOVA), $n = 16$ mice) and that reward probabilities at the time of leaving were not significantly different across trial types ($F_{(2,45)} = 0.049$, $p = 0.95$, one-way ANOVA, $n = 16$ mice). In contrast, in a shuffled data set in which trials were randomly assigned to one of the three trial types, poke numbers were, as expected, similar across trial types ($F_{(2,45)} = 0.046$, $p = 0.95$, one-way ANOVA, $n = 16$) and reward probabilities at the time of leaving varied with trial quality ($F_{(2,45)} = 4.13$, $p = 0.022$, one-way ANOVA, $n = 16$ mice; see Methods for details of the shuffling procedure). The comparison between shuffled and real data supports the idea that the mice were sensitive to the statistics of individual port visits and suggests that the mice employed a near-optimal strategy in this task.

**Activation of DRN neurons prolongs active exploitation.** We next examined the effect of activation of DRN 5-HT neurons in the probabilistic foraging task. Of the 16 mice examined, 10 expressed Cre-recombinase under the control of the SERT promoter (SERT-Cre) and 6 were wild-type littermates. Both groups of mice were infected in the DRN with a viral vector containing Cre-dependent channelrhodopsin-2 (AAV2/9-Dio-ChR2-eYFP) and implanted with an optical fiber cannula above the site of infection (Fig. 3a and Supplementary Fig. 2)[27]. After 9 days of training, we started a 10-day testing period, in which we photostimulated randomly on 50% of the correct trials. Stimulation was triggered by the first nose-poke in each trial and lasted until the end of the trial (Fig. 3b). We found that the number of nose-pokes in photostimulated trials was significantly greater than in control trials in ChR2-expressing mice ($14.73 \pm 0.55$ vs. $13.61 \pm 0.53$ pokes per trial (mean ± SEM); $p = 0.014$, Wilcoxon signed-rank test, $n = 10$ mice; see Supplementary Fig. 3 for individual mouse data). This difference was also significant comparing ChR2-expressing and wild-type mice ($1.12 \pm 0.36$ vs. $-0.46 \pm 0.18$ (mean ± SEM); $p = 0.0047$, Wilcoxon rank-sum test, $n = 10$ SERT-Cre and 6 wild-type mice; Fig. 3c–e). The effect of photostimulation was short-lived, occurring only on photostimulated trials without affecting behavior on subsequent trials (Fig. 3f).

**Activation of DRN neurons biases leaving but not travel time.** These results contradict the hypothesis that 5-HT promotes behavioral inhibition, as the willingness of mice to perform active nose-poking was enhanced rather than reduced by optogenetic activation of DRN 5-HT neurons. We hypothesized that this phenomenon could reflect a bias in the process underlying the decision of whether to stay and poke again or leave after each nose-poke. Consistent with this idea, we also found that photostimulation, which continued until the mouse exited the ROI, increased the delay to leave the foraging site, i.e., the interval

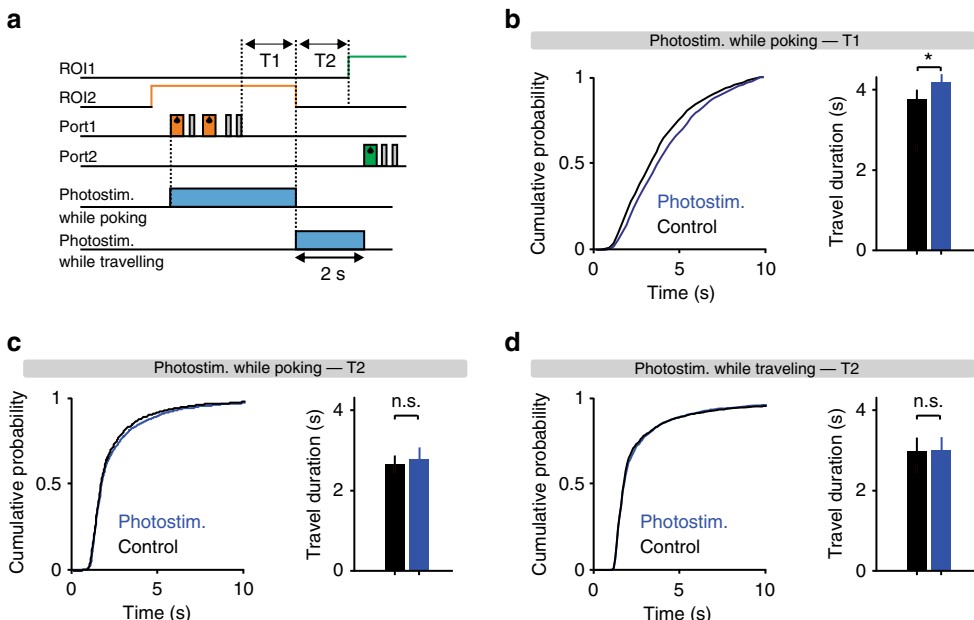

**Fig. 4** Lack of effect of DRN 5-HT photostimulation on travel duration. **a** Schematic diagram of task events during a single trial showing the period of photostimulation. In the first protocol (photostimulation while poking), photostimulation was triggered by the first poke in 50% of correct trials, and in the second protocol (photostimulation while traveling), photostimulation was triggered by ROI exit in 50% of the correct trials and lasted for 2 s. We also defined two relevant time periods: (T1) time-to-leave—the interval between the last poke and ROI exit, and (T2) travel—the interval between ROI exit and subsequent ROI entry. **b** Left: Cumulative distributions of T1 durations for photostimulated (blue) and control (black) trials, averaged across the population of SERT-Cre mice ($n = 10$) in the photostimulation-during-poking protocol. Right: Bar plot showing the corresponding averages. *$p < 0.05$, Wilcoxon signed-rank test. **c** Left: Cumulative distributions of T2 durations for photostimulated (blue) and control (black) trials, averaged across the population of SERT-Cre mice ($n = 10$) in the photostimulation-during-poking protocol. Right: Bar plot showing the corresponding averages. $p > 0.05$, Wilcoxon signed-rank test. **d** Left: Cumulative distributions of T2 durations for photostimulated (blue) and control (black) trials, averaged across the population of SERT-Cre mice ($n = 6$) in the photostimulation-during-traveling protocol. Right: Bar plot showing the corresponding averages. $p > 0.05$, Wilcoxon signed-rank test

between the last poke and ROI exit (control vs. stimulated trials: $3.84 \pm 0.25$ s in control vs. $4.26 \pm 0.23$ s in stimulated trials (mean ± SEM); $p = 0.0039$, Wilcoxon signed-rank test, $n = 10$ mice; Fig. 4b). In contrast, the time it took mice to travel from one port to the other side was not affected (control vs. stimulated trials: $2.64 \pm 0.23$ vs. $2.77 \pm 0.31$ s (mean ± SEM); $p = 0.56$, Wilcoxon signed-rank test, $n = 10$ mice; Fig. 4c). However, the fact that photostimulation was not delivered during the travel period might explain the lack of effect on this measure. Therefore, we subsequently tested 6 of the 10 ChR2-expressing mice from our original cohort using a protocol in which a 2-s stimulation was triggered by ROI exit, so that stimulation took place during the travel period (Fig. 4a). In this case as well, no effect of photostimulation on travel time was observed (control vs. stimulated trials: $3.02 \pm 0.31$ vs. $3.19 \pm 0.33$ s (mean ± SEM); $p = 0.22$, Wilcoxon signed-rank test, $n = 6$ mice; Fig. 4d).

**The proportional hazards model of probabilistic foraging.** Our analysis thus far suggests that DRN 5-HT neuron stimulation biases the decision process that sets the tradeoff between continued nose-poking ('exploitation') and leaving the site ('exploration'). We next sought to quantify this process more directly. Despite the agreement between our data and the predictions of the MVT (Fig. 2), we deemed it unlikely that the mice were using it directly for two main reasons: (1) the MVT requires an exact calculation of the expected reward probability after each nose-poke, which would require knowledge of the underlying task statistics, memory of the preceding sequence of rewards and omissions, and representation the three reward probabilities

corresponding to the three trial types (Eq. 2 in Methods); (2) the MVT is deterministic in nature and cannot account for the substantial variability we observed in mouse leaving times (this can be seen in the wide range of reward probabilities at the time of leaving in Fig. 2f; the MVT predicts these curves to be step functions)[28], [29]. Indeed, several simpler heuristics have been proposed to explain decision-making in probabilistic foraging tasks[30]. Notably, a model known as the "proportional hazards" model[31] not only aims at predicting mean leaving time but also models a stochastic decision process that leads to leaving[26], [32–34]. This model assumes that leaving decisions are taken randomly after each omission, based on an underlying hazard function (the conditional probability of leaving after $n + 1$ omissions, having experienced $n$ consecutive omissions) and that the hazard function resets after each reward, albeit at a different value, due to the influence of various predictor variables (Fig. 5a and Eq. 3, see Methods for further details of this model). We compared this model to a stochastic model based on the MVT's predictions (using the expected reward probability as the main predictor of leaving decisions) and found that the proportional hazards provided a better fit to the data in 15 out of the 16 mice tested. The main difference between the two models lies in the resetting property of the proportional hazards model. This feature results in a high win-stay probability, i.e., a high probability of staying after a reward, even for late rewards ("lucky" attempts that do not reflect the actual low reward probability), which was also evident in the data but not in the MVT model (Supplementary Fig. 4).

To test how well the proportional hazards model captures individual mouse behavior, we used half the data to fit model

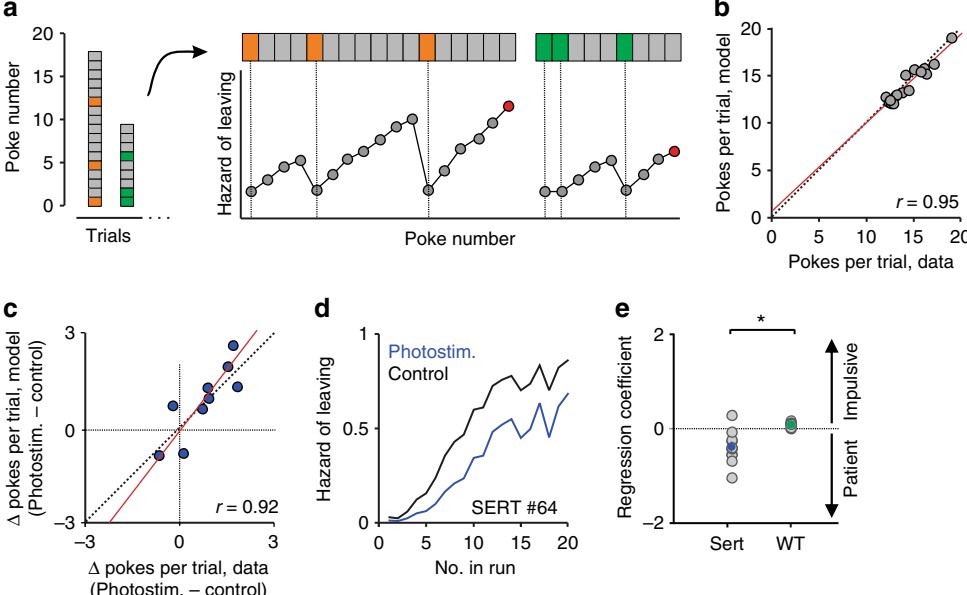

**Fig. 5** Fitting behavioral data using the proportional hazards model. **a** Schematic drawing of proportional hazards model fitting pipe-line for two example consecutive trials. Nose-pokes (pooled across trials and sessions for each mouse independently) were used to fit a logistic regression model—the outcome of which was an estimated hazard rate that is reset at trial start and after each reward and is multiplicatively changed by the different coefficients' values. This hazard rate can be viewed as an estimate of the probability of leaving after each nose-poke and can therefore be used to simulate mouse leaving decisions. Leaving decisions depicted as red circles. **b** Scatter plot of simulated vs. real average number of nose-pokes per trial. Each circle represents one mouse ($n = 16$). Dashed line is the unity diagonal and red line is a linear regression curve, with its correlation coefficient shown as well ($p < 0.001$). **c** Same as **b** for the effect of photostimulation on the number of nose-pokes per trial ($p < 0.001$). **d** The modeled hazard function for an example SERT-Cre mouse for phostostimulated and control nose-pokes. Note that decreased hazard means longer staying. **e** Cox regression photostimulation coefficient for SERT-Cre ($n = 10$) and wild-type ($n = 6$) mice. Averages across mice are shown in filled circles. *$p < 0.05$, Wilcoxon rank-sum test

parameters and cross-validated it using the second half. These simulations were in excellent agreement with our data (Supplementary Fig. 4). Furthermore, the model was accurate enough to predict idiosyncric individual differences in poke numbers across mice (Fig. 5b), as well as the effect of photostimulation on a mouse-by-mouse basis (Fig. 5c). When we examined the effect of photostimulation on the estimated hazard rate, expressed as the Cox regression coefficient for that variable, we found it to be significantly negative (that is, divisive or patience promoting) in SERT-Cre mice ($p = 0.014$, Wilcoxon signed-rank test, $n = 10$ mice) and significantly lower than that of wild-type mice ($p = 0.0075$, Wilcoxon rank-sum test, $n = 10$ SERT-Cre and 6 wild-type mice; Fig. 5e).

Next, since the proportional hazards model assumes that stimulation's effect is constant throughout trials (rather than changing as a function of duration), we tested whether this was indeed the case. To address this question, we first recorded multi-unit DRN responses in ChR2-expressing mice to 15 s long trains of optogenetic stimulation. We found that 5-HT neurons kept responding to this stimulation, which was comparable in both frequency and duration to the one used in our behavioral experiments (Supplementary Fig. 5). Since, according to the proportional hazards model, rewards reset the decision process, we reasoned that if the effectiveness of stimulation was changing with time, it should be different for trials in which last reward occurred early vs. late in the trial. However, Supplementary Fig. 6 shows that there was no correlation between the time of the last reward and the effect of photostimulation on the number of subsequent pokes before leaving, suggesting that the effect of stimulation on behavior was constant throughout its duration.

**Vigor of nose-poking reports the hazard rate of leaving.** The estimated hazard function in the proportional hazards model can be interpreted as a latent decision variable—the instantaneous propensity to switch foraging sites. This variable plays a similar role to that of accumulated evidence in integration-to-bound decision models[35]. In this interpretation, the effect of optogenetic activation of DRN 5-HT neurons is to slow down the dynamics of the decision variable. To try to test this more directly, we examined the correlation between different behavioral variables related to nose-poking and the latent decision variable in the proportional hazards model (i.e., the estimated hazard of leaving; Supplementary Fig. 7). We found that the duration of nose-pokes for omitted rewards and the duration of inter-poke intervals mirrored the value of the decision variable (Supplementary Fig. 7). Using these two variables, we calculated a measure that we refer to as the "omission duty cycle" (ODC: the ratio between each nose-poke's duration and the sum of the duration and the preceding inter-poke interval; Fig. 6a). The ODC, which essentially measures the rate or vigor of nose-poking, was indeed strongly negatively correlated with the model decision variable (Fig. 6b).

While both the decision variable of the proportional hazards model and the ODC change monotonically through a trial, this pattern is reversed when aligning on the last reward due to the resetting effect of rewards, such that the hazard is higher immediately before the last reward compared to immediately after it ($0.056 \pm 0.0041$ before and $0.014 \pm 0.0023$ after the last reward (mean ± SEM); $p = 0.00044$, Wilcoxon signed-rank test, $n = 16$ mice; Fig. 6c, d). We found that, as predicted, this reversal was true for the ODC as well ($0.86 \pm 0.010$ before and $0.87 \pm 0.011$ after the last reward (mean ± SEM); $p = 0.0084$, Wilcoxon signed-rank test, $n = 16$ mice; Fig. 6c, d). These results provide evidence

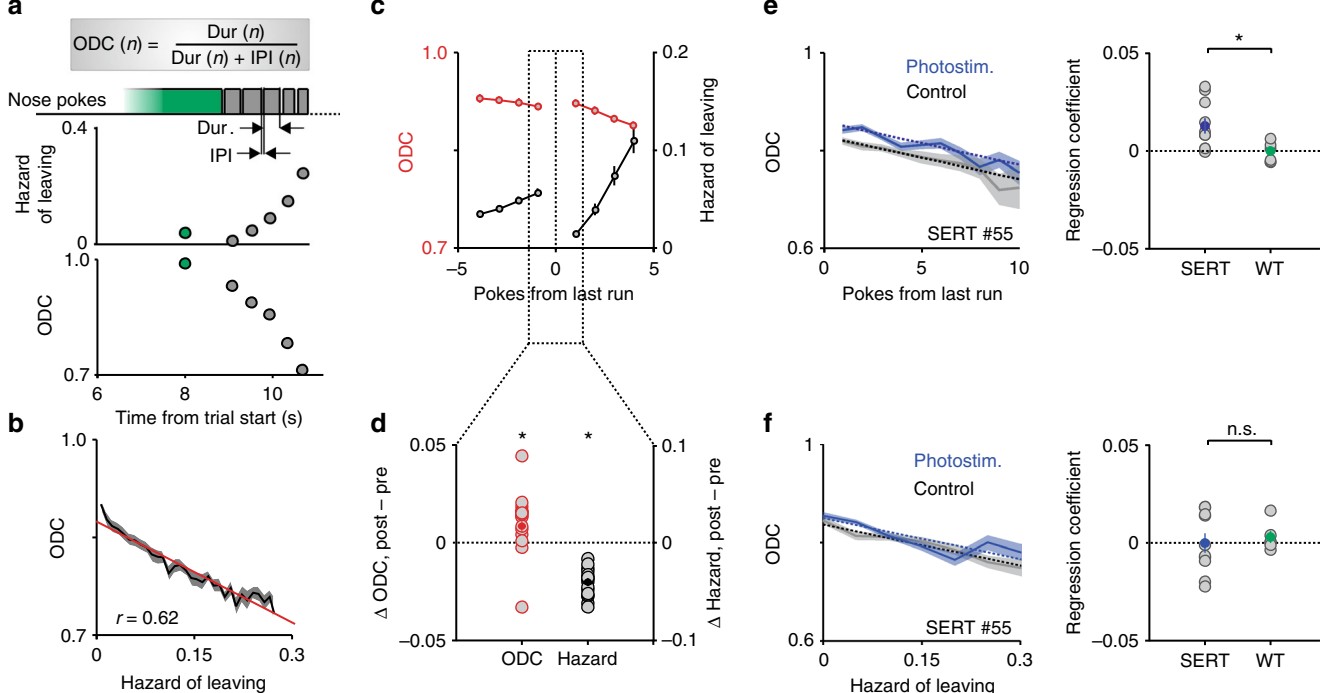

**Fig. 6** The effect of DRN 5-HT photostimulation on the microstructure of behavior. **a** Example trial represented in real time and starting from the last reward in that trial. Top: Each rectangle represents a single nose-poke (green marks the last reward and gray the subsequent omissions). Middle: estimated hazard for each nose-poke. Bottom: omission duty cycle (ODC). **b** Correlation between ODC and estimated hazard. The red line is a linear regression curve, with its correlation coefficient shown as well ($p < 0.001$). **c** Rewards reverse an overall (across trial) monotonic decrease in ODC (red) and an overall increase in hazard (black). The plot shows the average ODC and hazard as a function of poke number, aligned on the last reward and averaged across the population of mice ($n = 16$). Note that despite a decreasing (increasing) trend, the ODC (hazard) immediately after the last reward is higher (lower) than the one just before it. **d** Difference between average ODC and hazard immediately before the last reward compared to immediately after it across the population of mice ($n = 16$). *$p < 0.05$, Wilcoxon signed-rank test. **e** Left: The ODC for an example SERT-Cre mouse for photostimulated and control nose-pokes aligned on last reward. The dashed lines are linear regression curves ($p < 0.001$ for the photostimulation coefficient). Right: Regression photostimulation coefficients for SERT-Cre ($n = 10$) and wild-type ($n = 6$) mice. Averages across mice are shown in filled circles. *$p < 0.05$, Wilcoxon rank-sum test. **f** Left: The ODC for the same SERT-Cre mouse shown in **d** for phostostimulated and control nose-pokes as a function of estimated hazard. The dashed lines are linear regression curves, ($p > 0.05$ for the phostostimulation coefficient). Right: Regression photostimulation coefficients for SERT-Cre ($n = 10$) and wild type ($n = 6$) mice. Averages across mice are shown in filled circles. $p > 0.05$, Wilcoxon rank-sum test

that the vigor with which mice nose-poke is: (1) monotonically decreasing in a manner reflecting the decreasing reward probability; and (2) non-trivially correlated with the effect of rewards on the modeled decision variable. The latter observation argues against the possibility that simpler variables that would be expected to evolve monotonically during nose-poking, such as muscle fatigue, could account for the observed relationships.

Finally, we examined the effect of DRN 5-HT neuron photostimulation on ODC. Since stimulation decreases the hazard of leaving (Fig. 5) and ODC is negatively correlated with the hazard rate, we predicted that the ODC would be higher in photostimulated vs. control trials. Using linear regression analysis, we found that photostimulation indeed increased the ODC of SERT-Cre mice ($p = 0.0098$, Wilcoxon signed-rank test, $n = 10$; see Supplementary Fig. 8 for individual mouse data) and that this increase was significantly higher in SERT-Cre compared to wild-type mice ($p = 0.011$, Wilcoxon rank-sum test, $n = 10$ SERT-Cre and 6 wild-type mice; Fig. 6e). Furthermore, we reasoned that, if the ODC truly reflects the modeled hazard function, then it ought to be the same in photostimulated vs. control trials, when conditioned on the hazard function, since this quantity already takes into account the effect of photostimulation on the decision process. Using similar linear regression analysis, we found that, when plotting the ODC as a function of the modeled hazard, photostimulation had no effect on the ODC of

SERT-Cre mice ($p = 0.90$, Wilcoxon signed-rank test, $n = 10$) nor was there a difference between SERT-Cre compared to wild-type mice ($p = 0.54$, Wilcoxon rank-sum test; Fig. 6f).

These results show that the vigor of poking behavior is closely linked to the underlying decision-making process and can be used as a real-time read-out of a latent decision variable[36]. Furthermore, the effects of DRN 5-HT neuron stimulation on foraging behavior were well explained by a model in which it biases decisions by multiplicatively scaling the latent decision variable on which leaving decisions are based.

## Discussion

In this paper, we found that optogenetic activation of DRN 5-HT neurons during foraging promotes active exploitation of the current foraging site, increasing the time to switch sites. This result in some ways resembles previous findings that DRN 5-HT neuron activation increases the ability to wait for delayed outcomes[15, 16]. However, because in this task "waiting" required active behavior (nose-poking) the results argue strongly against the hypothesis that 5-HT enhances waiting by enhancing behavioral inhibition. Rather, 5-HT appears to stabilize an on-going behavior by reducing the probability of switching to the next one, i.e., stabilizing nose-poking while reducing the probability of leaving. This behavior could be well described by a simple model,

the proportional hazards model[26, 32–34]. In this model, the effect of 5-HT stimulation was to proportionally scale down the hazard rate, thereby reducing the probability to leave. Thus 5-HT appears to act as multiplicative bias on the latent decision variable in the foraging decision process.

Foraging behavior is fundamental to any animal's survival. Natural environments require animals to balance various consideration in order to achieve optimal results[37]. Of particular interest is the exploration–exploitation dilemma: should the animal stay and exploit a depleting resource, or leave instead to forage elsewhere? Within this framework, the MVT describes the optimal strategy, which is deciding to leave at the point in time when the instantaneous rate of rewards drops below its average value[25]. However, the MVT, as originally formulated, applies to deterministic environments, where foragers have perfect knowledge of the instantaneous and average reward rates. Although these assumptions are not realistic, and the solution for the stochastic case is quite intricate[28], a key prediction of optimality remains relevant: switching decisions should be made when instantaneous reward probability is equal to some fixed value, irrespective of whatever has happened up to this point[26, 33]. Our study confirms that mice follow this prediction.

In order to model the dynamics of the stochastic environment, we used a simple heuristic decision rule, the proportional hazards model[31]. We found that this model captures not only the fixed relationship between instantaneous and average reward rate at leaving time but also the variability of this decision point across the mouse population (Fig. 5b) and the case-by-case effects of photostimulation across mice (Fig. 5c). Here the effect of DRN 5-HT neuron activation by photostimulation was to decrease the hazard rate in a proportional manner. It thereby increased the average time to reach decision threshold, increasing the time spent nose-poking before deciding to leave.

The proportional hazards model shares commonalities with models that have been widely used in related fields, such as interval timing[38] and perceptual decision-making[35]. In this class of models, before reaching a decision, agents integrate information in the form of elapsed time[39, 40] or noisy samples of sensory input[35, 41]. The amount of accumulated information during the accumulation process is a latent or hidden variable that, through behavioral studies, can only be indirectly inferred through modeling the timing of overt behavior: the timing and type of choices made. An important source of evidence in favor of such models has been provided by the ability to find correlates of decision variables in neural activity patterns recorded in brain regions hypothesized to underlie the decision process[35, 41, 42]. Here we found that a dynamic and readily observable microfeature of mouse behavior, the vigor of nose-poking during foraging, strongly correlated with the latent decision variable in the proportional hazards model. This observation offers support for the validity of the proportional hazards model for foraging. It also supports the notion that readily observable dynamic microfeatures of behavior reflect latent decision variables, providing general insight into the "covert" processes underling "overt" decisions[43, 44].

Our success in modeling the effect of DRN 5-HT neuron stimulation using a simple decision model encourages us that a computationally based and behaviorally constrained approach will be a valuable way to approach the interpretation of 5-HT function. At the same time, our results leave open the question of how more precisely to interpret the influence of 5-HT, since the decision variable that was modulated is highly abstract and represents the conjunction of many more specific factors that might enter into the decision to stay or leave. Broadly speaking, these fall into two categories. First, there are factors that represent the calculation of the costs and benefits of decision options. These

include the cost or effort of foraging at a given site ("handling cost" in the foraging literature), the expected reward gained for each foraging attempt, and the expected cost of travel to the next foraging site. Second, there are factors related to the uncertainties inherent in the probabilistic foraging task, especially the uncertainty related to the stochastic nature of reward delivery[37].

For an optimal Bayesian agent, estimating the costs and benefits of decision options must take into account the uncertainties of those estimates. Thus the impact of DRN 5-HT neuron stimulation on the hazard rate of leaving might be due to a more direct effect on costs or benefits of staying vs. leaving or it might arise from a modulation of uncertainty which only indirectly impacts estimates of cost and benefit[45]. We currently favor the latter interpretation for two reasons. First, the endogenous activity of 5-HT neurons reports unexpected events regardless of valence, as appropriate for a signal related to uncertainty[46]. Second, a growing set of evidence suggests that 5-HT does not exert the sort of motivational biases that arise from natural rewards or punishments[16, 47]. If 5-HT modulated the costs or benefits of on-going events with which it was correlated, one might expect this to be revealed in a broad range of situations. Yet, 5-HT stimulation does not produce place preference or bias thigmotaxis in the open field[48] and 5-HT stimulation applied during the outcome of a two-alternative choice value-based decision-task does not bias choices[16]. While these results are at odds with those reported by Liu et al.[49], which had found a reinforcing effect of optogenetic DRN 5-HT neuron stimulation on behavior, differences in the transgenic mouse lines used to drive ChR2 expression (ePet1-Cre in Liu et al. vs. SERT-Cre here) or the targeting of different anatomical parts of the DRN may account for this apparent discrepancy[47]. In contrast, elevated 5-HT levels have also been associated with aversive processing[50, 51]. For example, acute tryptophan depletion (an experimental manipulation used to lower 5-HT levels in humans) attenuates punishment-induced inhibition[52], suggesting that endogenous 5-HT release may reduce response vigor in the face of aversive predictions[53]. However, these effects were shown to be specific to aversive contexts, which were absent in the current study. Possible context-dependent differences in 5-HT's effect on foraging behavior could be tested in future experiments by, for example, introducing probabilistic punishments during resource exploitation, and testing the effect of DRN 5-HT stimulation on response vigor under these conditions.

One limitation of the present study is that the task was too complex to be modeled precisely by a simple model; the proportional hazards model is not the optimal solution to this task and the complexity of the optimal solution suggests that it is not likely used by mice. This issue could be addressed in future studies using refined tasks in which simple optimal solutions exist. This, together with additional manipulations of task variables, should help to ascertain more precise interpretation of the influence of 5-HT.

Here we have shown that optogenetic activation of DRN 5-HT neurons prolongs the willingness of mice to forage for stochastically delivered rewards. Waiting can be considered a foraging situation in which rewards are expected but not forthcoming and the same class of integrate-to-threshold models can be used to model both foraging and waiting tasks[38–40]. On the other hand, in waiting tasks, passivity and waiting are coincident, whereas in foraging tasks, waiting and passivity are decoupled: the choice is no longer between passivity and activity but between two different types of active behavior, nose-poking, or locomoting. The reason that 5-HT stimulation favors patient waiting is apparently not because it favors behavioral inhibition or passivity but because it favors persistence in a current behavior, even if it is active. One possible interpretation of this finding is that 5-HT it

favors persistence by making behavioral transitions (switching from one behavior to another) less likely. This interpretation can explain the observation that DRN photostimulation does not bias travel times between reward sites when stimulation occurs after the mouse is already in transit (Fig. 4 and ref. [48]). It would also be consistent with an observed increase in active escape behavior in the forced swim test that is induced by stimulation of medial prefrontal cortex axons in the DRN[54] and enhanced lose-shift behavior in probabilistic reversal tasks, observed in patients with major depression[55] and in individuals homozygous for the long 5-HT transporter allele[56], both associated with decreased levels of extrasynaptic 5-HT. However, as discussed above, another interpretation is that decreased behavioral transitioning is the consequence of a change in one or more underlying decision variables. For example, by increasing the perceived uncertainty of the instantaneous estimate of the reward rate, 5-HT could bias decisions in favor of continued nose-poking in the task[45].

Despite the tantalizing consistency of the above results, one should not lose site of the complexity of the 5-HT literature, which implicates this molecule in many functions with little obvious relationship to foraging, waiting, or persistence. However, the application of relatively abstract computational models may provide a useful way toward a more satisfactory overarching understanding. Rather than seeing 5-HT's effects in terms of a suite of diverse behaviors for which the overarching behavioral theme is missing, one may look to commonality of action at the algorithmic level. If 5-HT has a common action on circuits that implement integration-to-bound, then the range of behaviors in which 5-HT acts, such as waiting[13], cognitive flexibility[46, 57], and sensorimotor gain control[58], will be as diverse as the instances in which this algorithm is applied. For example, in the nematode *Caenorhabditis elegans* it was found that 5-HT favors dwelling in a food patch over roaming[59], and in larval zebra fish 5-HT is implicated in modulation of short-term locomotor memory to change visuo-motor gain[60]. While the relationship between sensorimotor gain adaptation and probabilistic foraging is obscure at the behavioral level, neural integration is a common component of both. The investigation of 5-HT action at the level of postsynaptic target circuits, together with computational modeling, will be important to test such hypotheses.

## Methods

**Animal subjects**. Mouse lines, surgical procedures for virus injections and optic fiber cannula implantation, and optical set-ups for optogenetic stimulation were identical to those described in previous papers from our laboratory[27].

Sixteen adult male C57BL/6 mice (10 SERT-Cre mice and 6 wild-type littermates) were used in this study. All experimental procedures were approved and performed in accordance with the Champalimaud Centre for the Unknown Ethics Committee guidelines and by the Portuguese Veterinary General Board (Direcção-Geral de Veterinária, approval 0421/000/000/2016). The SERT-Cre mouse line[61] was obtained from the Mutant Mouse Regional Resource Centers (stock number: 017260-UCD). The mice were kept under a normal 12 h light/dark cycle, and training as well as testing occurred during the light period. During training and testing the mice were water deprived, and water was available to them only during task performance. Food was freely accessible to the mice in their home cages.

Behavioral training started 2–10 weeks after virus injection and lasted for 9 days, at the end of which we commenced testing (experimentation). Testing periods consisted of 10 consecutive daily sessions. Ten of the 16 mice were only tested once, and 6 SERT-Cre mice (out of the 10) were tested again using a different photostimulation protocol (see below). This second testing period started 1 month after the end of the first one. The experimenters were blind to the mice's genotype throughout training and testing periods.

**Adeno-associated virus injection and cannula implantation**. Mice were anesthetized with isoflurane (4% induction and 0.5–1% for maintenance) and placed in a stereotaxic frame (David Kopf Instruments, Tujunga, CA). Lidocaine (2%) was injected subcutaneously before incising the scalp. In order to infect serotonergic neurons with ChR2, a craniotomy was drilled over the cerebellum and a pipette

filled with a viral solution (AAV2.9.EF1a.DIO.hChR2(H134R)-eYFP.WPRE.hGH, $10^{13}$ GC mL$^{-1}$, University of Pennsylvania) was lowered to the DRN (Bregma −4.7 AP, −2.9 DV) with a 32° angle toward the back of the animal. The viral solution (1 μL) was injected using a Picospritzer II (Parker). After waiting for 10–15 min, the pipette was removed from the brain and an optical fiber (200 μm core diameter, 0.48 NA, 4–5 mm long, Doric lenses) was lowered through the same craniotomy such that its tip was positioned 200 μm above the injection point. The implant was cemented to the skull using dental acrylic (Pi-Ku-Plast HP 36, Bredent, Senden, Germany). Mice were monitored until recovery from the surgery and returned to their home cages. Gentamicin (48760, Sigma-Aldrich, St. Louis, MO) was topically applied around the implant.

**Optogenetic stimulation**. In order to optically stimulate ChR2-expressing 5-HT neurons, we used blue light from a 473 nm laser (LRS-0473-PFF-00800-03, Laserglow Technologies, Toronto, Canada or DHOM-M-473–200, UltraLasers, Inc., Newmarket, Canada) that was controlled by an acousto-optical modulator (AOM; MTS110-A1-VIS or MTS110-A3- VIS, AA optoelectronic, Orsay, France). Light exiting the AOM was focused into an optical fiber patchcord (200 μm, 0.22 NA, Doric lenses), connected to a second fiber patchcord through a rotary joint (FRJ 1 × 1, Doric lenses), which was then connected to the chronically implanted optic fiber cannula.

**Histology**. In order to confirm successful viral expression of ChR2-eYFP and optical fiber placements, we used postmortem histology at the end of the experiments. Mice were deeply anesthetized with pentobarbital (Eutasil, CEVA Sante Animale, Libourne, France) and perfused transcardially with 4% paraformaldehyde (P6148, Sigma-Aldrich). The brain was removed from the skull, stored in 4% paraformaldehyde overnight, and kept in cryoprotectant solution (PBS in 30% sucrose) for 1 week. Sagittal sections (50 mm) were cut in a cryostat (CM3050S, Leica, Germany) and mounted on glass slides with mowiol mounting medium (81381, Sigma-Aldrich, St. Louis, MO). Scanning images for yellow fluorescent protein (YFP), and transmitted light were acquired with an upright fluorescence microscope (Axio Imager M2, Zeiss, Oberkochen, Germany) equipped with a digital CCD camera (AxioCam MRm, Zeiss) with a 5× or 20× objective. In a previous study using the same Cre-dependent optogenetic approach and the same mouse line, we reported that 94% of ChR2-eYFP-positive neurons were serotonergic[27].

In the analysis shown in Supplementary Fig. 2, we examined the correlation between DRN 5-HT photostimulation and fiber position or ChR2 expression levels. To do so, we first determined, for each mouse, the fiber's distance from the midline by observing which of the sagittal slices contained the most fiber damage. Next, we counted the number of cells in that slice using the ImageJ software's (imagej.net) cell counter plugin.

**Electrophysiology**. In order to confirm the effectiveness of our photostimulation protocol, we recorded electrophysiological responses in ChR2-eYFP-expressing anesthetized mice (Supplementary Fig. 5). An optrode consisting of an optical fiber (200 μm diameter) coupled to a 470 nm laser (Laserglow Technologies) and a microelectrode (1–3 MΩ; FHC) was lowered into the DRN at a 32° angle. Multi-units were acquired digitally using the Spike2 software (Cambridge Electronic Design). Data were stored on a personal computer for offline analysis. Each recording session consisted of 10 15 s long photostimulation sweeps (10 ms pulse width, 25 Hz) and a 60 s interval was inserted between sweeps to allow for activity to return to baseline. In total, we recorded seven multi-units from two ChR2-expressing SERT-Cre mice.

**The probabilistic foraging task**. Sixteen water-deprived mice were trained in a probabilistic foraging task. The apparatus was an elongated $50 \times 14$ cm$^2$ chamber with two water-reward ports, one at each end. Each port had an infrared emitter/sensor pair located on its sides to measure nose-pokes (model 007120.0002, Island motion corporation) and a metallic tube running through its center for water delivery (controlled by a valve—LHDA1233115A, The Lee Company, Westbrook, CT). Four partitions were placed at regular intervals along the chamber. Each partition blocked about one half of the corridor's width, forcing the mice to zig-zag between them when crossing from side to side. All task-related events were controlled using a behavioral control system, Bcontrol, that was developed by Carlos Brody (Princeton University) in collaboration with Calin Culianu, Tony Zador (Cold Spring Harbor Laboratory), and Z.F.M.

Each trial consisted of a sequence of nose-pokes in one of the two reward ports. Each nose-poke was rewarded with some probability by a 3 μl drop of water. Reward probabilities decreased after each nose-poke, forcing the mice to alternate between sides during the session. Correct trials were ones in which the mice alternated sides, and error trials were ones in which the mice returned to the same reward port without nose-poking in the other one. Reward probabilities in each correct trial were drawn from one of the three, equally likely exponentials, each

decreasing with poke number

$$P(o_n = 1|t_i) = A_i e^{\frac{-(n-1)}{5}} \qquad (1)$$

where $t_i$ is the $i$th trial type ($i = 1,2,3$) corresponding to low-, medium-, and high-quality trials. These types differed in their exponential scaling factors, such that $A_1 = 0.5$, $A_2 = 0.75$, $A_3 = 1$. $N$ marks the poke number within a trial, and $o_n$ is the outcome of the $n$th poke (1 for reward and 0 for omission). Trial types (exponential scaling factors) were randomly interleaved and trial-type identity was not cued to the mice. Reward probability was set to zero during error trials. For technical reasons, the reward probability was set to zero after the 20th nose-poke; this led to only a marginal deviation from true exponentials.

The mice were tracked on-line using Bonsai, a visual programming framework[62], and their position was used to control trial transitions. We defined a 10 cm square ROI around each port. ROI exits signaled the end of one trial and the beginning of the next one. Additionally, to avoid exceptionally long photostimulation durations, trials were terminated if 10 s had elapsed since the last nose-poke exit, even if the mouse was still in the ROI.

Each testing session lasted about 30 min in which the mice performed at least 80 trials, gaining an average of 3.75 rewards per trial. We found that the behavior of the mice was very consistent throughout the session. However, some trials, particularly towards the end of the session, were unusually short, thus making the distribution of poke numbers per trial bimodal (Supplementary Fig. 1). We therefore consider for analysis only the first 60 trials in each session and only those trials that contained more than two pokes in them.

**Photostimulation protocols**. In this paper, we present the results of two experiments. The behavioral task was identical in both (as described above), yet the photostimulation protocol differed between the two. In the first experiment, photostimulation (a train of 10 ms wide pulses at 25 Hz, measuring 5 mW at fiber tip) was triggered by the first nose-poke in a trial and lasted until the trial's end (i.e., by either ROI exit, or if 10 s had elapsed since the last nose-poke exit). In the second experiment, a similar photostimulation train was triggered by ROI exit and lasted for 2 s. Sixteen Blue LEDs were placed inside the box, eight above each port, and delivered a flickering masking light, to prevent behavioral changes due to the mere laser light flashes during photostimulated trials. The masking light was identical to the photostimulation in frequency and duration and was present in all trials.

**Data analysis**. All data analysis was performed using the custom-written software in MATLAB (Mathworks, Natick, MA). When comparing between groups, we used either one-way ANOVA for larger sample sizes ($n = 16$) that met ANOVA assumptions (vartestn and lillietest MATLAB functions testing for homogeneity of variance and normality, respectively) or non-parametric Wilcoxon tests for smaller sample sizes. Linear regression analysis was used to test significant effects of continuous variables. In all figures, average data and error bars, or shaded patches around curves, represent mean ± SEM.

**Subjective reward probability**. In the analysis shown in Fig. 2, we calculated the reward probability at the time of leaving (i.e., the expected reward probability in the next poke, if it were to be made). While the actual probabilities are given by Eq. 1, trial types were not cued to the mice, making it impossible to know exactly each poke's reward probability. Instead, we considered the most accurate estimate, assuming perfect knowledge of the task's structure and reward history leading up to a nose-poke. The probability of gaining a reward in poke $n + 1$ given trial history $o_1, \ldots, o_n$ is thus:

$$P(o_{n+1} = 1|o_1, \ldots, o_n) = \sum_{i=1}^{3} P(o_{n+1} = 1|t_i) P(t_i|o_1, \ldots, o_n)$$
$$= \sum_{i=1}^{3} P(o_{n+1} = 1|t_i) \frac{P(o_1, \ldots, o_n|t_i)}{\sum_{j=1}^{3} P(o_1, \ldots, o_n|t_j)} \qquad (2)$$

Where $t_i$ is the $i$th trial type ($i = 1, 2, 3$) and $P(o_1, \ldots, o_n|t_i) = \prod_{j=1}^{n} P(o_j|t_i)$ is calculated using Eq. 1. In Fig. 2d, we use this quantity to obtain the instantaneous reward rate by calculating, for each nose-poke, the ratio between this and the poke's duration.

**Shuffled behavioral data**. In Fig. 2, we also compare real to shuffled data sets. Shuffling was done individually for each mouse by first considering all potential trials, namely, the 20 poke long sequences of rewards and misses that are calculated using Eq. 1. Next, we paired each such potential trial with a randomly chosen trial length, selected from the actual data. This resulted in a shuffled data set in which average trial lengths were preserved while all other correlations between trial reward structure (such as trial type) and behavior were destroyed.

**Travel time analysis**. In Fig. 4, we analyze the effect of photostimulation on travel times. Since the travel time is not well defined in error trials (when the mice leave and return to the same location without visiting the other reward port in-between), we considered for this analysis only those trials in which both current and subsequent trials were correct. Additionally, since photostimulation terminated if 10 s had elapsed since the last nose-poke exit, even if the mouse was still in the ROI, such trials were excluded from this analysis as well. Consistent with the result shown in Fig. 4b, the fraction of such trials was higher in the photostimulation condition (control vs. stimulated trials: 8.30 ± 2.49% in control vs. 13.30 ± 3.32% in stimulated trials (mean ± SEM); $p = 0.002$, Wilcoxon signed-rank test, $n = 10$ mice).

**Cox proportional hazards regression model**. In order to model individual mouse choice behavior and its modulation by various factors (e.g., DRN photostimulation), we used the Cox proportional hazards regression model. This semi-parametric model calculates the probability of leaving after each poke according to a baseline hazard (the probability of leaving immediately after a poke as a function of its number) that is estimated from the data and that may be changed multiplicatively by trial-general covariates. The model is reset after each reward but potentially to a different value, depending on the covariates. Therefore, we fitted leaving probabilities for all nose-pokes as a function of their distance from the previous reward rather that from trial start. To do so, each trial was segmented into one or more sub-trials or runs. Each such run was bracketed from the left by a reward (except for the first one, in cases where the first poke in a trial was an omission) and from the right (except for the last one, in the very common cases where the last poke in a trial was an omission). Note that length zero runs were also allowed, if two rewards were delivered consecutively. As noted in ref. [31], in the discrete case this model is equivalent to a logistic model of the form:

$$\frac{\lambda(n)}{1 - \lambda(n)} = e^{\beta x} \frac{\lambda_0(n)}{1 - \lambda_0(n)} \qquad (3)$$

where $\lambda(n)$ represents a hazard function (hazard rate of leaving after the $n$th poke. $\lambda_0(n)$ represents a baseline hazard function, that is, the hazard function when all the predictors are equal to 0, $\beta$ is a row vector with scalar Cox coefficients for each of the covariates, and $x$ is a column vector representing covariate values. The covariates we used were the position of the previous reward, the port side (this nuisance variable was irrelevant for our interpretation of the data but nonetheless was necessary for accurate fitting), and photostimulation condition (1 in stimulated trials and 0 otherwise).

We also used the fitted model parameters to simulate data. Similarly to the shuffled simulation described above, we first gathered all potential trials from the data. Next, to simulate leaving decisions we used the fitted Cox model to generate a sequence of probabilistic "coin-flips", such that if one assumes "heads" to represent the hazard of leaving after poke $n$ (taking into consideration its distance from the last reward, photstimulation condition, etc.), then for a given trial the simulation would go on with a series of coin-flips until the first "heads" is encountered—this would mark the time of leaving. In these simulations, half of the data were used for fitting the model and the second half, for testing.

To provide further support for our use of the proportional hazards model, we conducted model comparison analysis (Supplementary fig. 4). In it, we compared the proportional hazards model to an alternative model, based on the normative predictions of the MVT and Eq. 2. The model was formulated as follows:

$$\frac{\lambda(n)}{1 - \lambda(n)} = e^{\beta_0 + \beta_1 * P_{rew}(n+1) + \beta_2 * s} \qquad (4)$$

where $\lambda(n)$ represents the probability of switching sides after the $n$th poke. $P_{rew}(n)$ is the $n$th poke reward probability (calculated using Eq. 2), and $s$ is an indicator variable for side.

**Code availability**. All the original code used for data analysis is available upon request from Z.F.M. (zmainen@neuro.fchampalimaud.org).

**Data availability**. All data are available upon request from Z.F.M. (zmainen@neuro.fchampalimaud.org).

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

## Acknowledgements

We thank Bassam Atallah and Madalena Fonseca for comments on a previous version of the manuscript. We also thank Gil Costa for support with visual the diagram shown in Fig. 1. This work was supported by the European Research Council (Advanced Investigator Grants 250334 and 671251 to Z.F.M.) and Champalimaud Foundation (Z.F.M.).

## Author contributions

E.L., D.B., and Z.F.M. designed the experiments. D.B., E.L., P.V., D.S., and M.o.L conducted the experiments. E.L. and P.V. analyzed the data. E.L. and Z.F.M. wrote the manuscript.

## Additional information

**Competing interests:** The authors declare no competing interests.

