## [Peer Review File · Nature Communications]

Reviewers' comments:

Reviewer #1 (Remarks to the Author):

Lottem et al. manipulated 5-HT neurons in mice performing a foraging task. This is a nice manuscript, well motivated theoretically, and very timely.

Major comments

1. My main concern is the nagging question of whether the experiment is doing what the authors claim it is. Specifically, what are the effects of 5-HT neuron optogenetic stimulation? Are 5-HT neurons able to follow 25 Hz stimulation for the applied durations? It seems likely that many of the neurons could follow 25 Hz stimulation briefly, followed by a prolonged inhibition (presumably due to autoreceptors). Thus, the observed effects could actually be due to pauses of 5-HT neuron firing (or, possibly, biphasic responses of excitation followed by inhibition). This could be resolved by recording from 5-HT neurons while stimulating them, ideally in the awake mouse, but possibly in the anesthetized mouse.
2. How do the present results relate to those recently published from the Mainen lab (Correia et al., eLife, 2017)? Specifically, I wonder whether there are long-term effects of stimulation that can be dissociated from the short-term ones. For example, are pokes per trial different from control when there is a long sequence of consecutive trials with stimulation, immediately following such a sequence? A long time after such a sequence?
3. Please include more details of the stimulation (e.g., a histogram of stimulation durations).
4. Are there systematic differences between mice as a function of the amount or extent of ChR2 expression? It would be nice to see more detailed analysis of individual mouse behavior. Related to this, Fig. 1 would benefit from an entire example session (e.g., in the format of 1b or 1d).

Minor comments

1. I suggest replacing bar plots with histograms or box plots (or something that better represents the distributions). Also, Fig. 2b doesn't appear to have error bars.
2. Are there enough trials to examine the effects of stimulation on errors?
3. The data may not warrant a title that includes "exploitation," given the subtleties of exploration-exploitation balance, and the extent to which that was tested here.
4. Fig. 6 would be clearer if panels (c) and (d) had consistent coloring.
5. The model is confusing as written in Fig. 5a. Do the authors mean "logit(h_0)" instead of " h_0 "? Why use "h" instead of "lambda" here?

6. It would be useful to see, in addition to the subjective reward probability in Fig. 2g, the calculation based on Eq. 1. Similarly, in Fig. 2d, it would be useful to see actual reward rate.

Reviewer #2 (Remarks to the Author):

Summary

The authors trained 16 mice to perform a novel probabilistic foraging task. 10 of mice expressed Cre under the SERT promoter and other 6 were wild-type litter-mates. All animals had a cre-dependent channel rhodopsin injected into the dorsal raphe nucleus. The task required mice to run back and forth along a linear maze to ports on either end. Pokes into a port resulted in a probabilistic water reward, with the probability decaying exponentially with each poke. Amazingly, mice are near optimal at this task: they leave each port (on average) when the instantaneous probability of reward is equal to their average reward rate. Stimulation of the 5-HT neurons in the dorsal raphe increase the # of pokes the subjects make before leaving. The behavior, and the stimulation effects, seemed well-fit by a proportional hazards model. The main claim is that the work refutes the hypothesis that 5-HT activation leads to behavioral inhibition.

Overall, I found the paper clearly written and interesting. Taken at face value, the paper would result in a significant shift in the putative role of 5-HT: from "behavioral inhibition" to something more like a "persistence" signal (but maybe more mechanistic, by increasing the time-constants of accumulation to bound processes in general). Of course, a single experiment rarely can (or should) result in a major shift in the field, but I think that others will be inspired to further test this alternative view.

The Mainen lab has been working for some time on neural mechanisms related to "decisions to wait", and I think this paper is a nice step that I imagine (and hope) will lead to linking M2 recording with 5-HT stimulation. The authors posted this paper on Biorxiv which allowed me to discuss the paper with colleagues before the submission of the review. I recommend the paper for publication after my concerns are addressed.

Sincerely,
Jeffrey Erlich

Major Comments

1) Histology (and comparing with variation across animals)

The authors describe their histology in the methods, but did not show the results. It seems two factors observable in the histology might have explained the variation seen in the effects of stimulation across animals. (a) Overall levels of expression, (B) location of the fiber relative to the expression. Are either of these the case?

2) Alternative models

The authors acknowledge in their discussion that the cox model is hard to interpret. I think they do an excellent job in the discussion exploring the possibilities. However, it seems (maybe naively) that further modeling could rule out some potential confounds. For example, the authors claim that 5-HT neuron stimulation is not rewarding and doesn't generate conditioned place-preference. However that is directly contradicted by Liu et al 2014 (<http://dx.doi.org/10.1016/j.neuron.2014.02.010>). I was surprised that the authors failed to cite that paper. If 5-HT neurons (via 5-HT and glutamate) were directly rewarding what would the authors expect the result to be in the probabilistic foraging task? I imagine if it was strongly rewarding then the effect would be much bigger (e.g. animals would just hang out and enjoy the stimulation), but maybe the 5-HT neurons decrease their activity over the long stimulation (something we could see with simultaneous electrophysiology). Can this interpretation be ruled out? Without knowing the effect of the long stimulation duration on the 5-HT neurons it seems that it is quite difficult. One further experiment that would (i think) be very clear is to randomly stimulate on certain pokes rather than during the entire trial. If the stimulation is directly rewarding, it would act like a reset on the run (e.g. like water delivery). But if it just slowed the accumulation of decision-to-leave variable (as the authors claim) then it would have a smaller effect than the whole trial stimulation and no specific effect on poke-microarchitecture.

Minor Comments:

1) Reliability of the effects

It seems that the stimulation worked in 7/10 animals. Bar graphs that combine all data across animals do not give readers a good sense of the reliability of behaviors or effects. I would like to see most figures generated for each animal (can be supplemental figures). I would also recommend to the authors to strengthen the sensitivity of their statistics by using mixed-effects models (eg. for fig 2f) instead of ANOVA.

2) Model comparison.

The authors did not perform standard model comparison techniques, like leaving out parameters and checking AIC, BIC, DIC or MDL (minimum descriptive length). In others words, they showed one model that seemed to fit well, but didn't try any other models or show which parameters were "necessary".

3) on line 385, in the discussion, it would be nice for the authors to help a less experienced reader out by mentioning some of the other major roles of 5-HT with some cited reviews: e.g. depression, sleep, digestion.

4) Has anyone previously shown that mice are capable of near optimal foraging? The authors might want to emphasize that aspect of the results a bit more.

5) PFT is not a standard acronym, i would just find/replace with probabilistic foraging task. or in many cases you can just say "the task". Sometimes you say "the PFT" other times you omit the "the".

6) Difference between 5-HT neurons and 5-HT. The authors are stimulating 5-HT neurons but those neurons express glutamate as well. They authors should be more careful in their use of 5-HT vs. 5-HT neurons in the DRN.

7) Was there an increase in errors (return to the just visited poke) after stimulation? If so, that would support a reinforcing effect of the stimulation.

8) Was there any relationship between the individual variation in baseline behavior and effect of stimulation? It would be nice to see figure 2d, but with a second blue dot for the 10 SERT animals under stimulation. According to the "main effect" the blue dots should shift down relative to the black dots. Is this the case? I guess it might be very noisy.

Reviewer #3 (Remarks to the Author):

This study addresses the long-standing question of the role of serotonin in motivated decision-making. Specifically, the authors use a foraging task in combination with optogenetic activation of 5HT neurons to assess the willingness of mice to explore a depleting reward site, where they aim to pit against each other hypotheses of serotonin's role being characterised as behavioural inhibition (putatively indexed by leaving a patch early) versus patience / persistence. They show that the mice's persist longer in a depleting patch when their serotonin neurons are stimulated. I believe that this is an interesting finding and the role of serotonin in decision-making is indeed an important question. Optogenetic approaches allow for a new level of neurochemical specificity which particularly serotonin research has been lacking.

The authors' main claim is that (line 340) "our results [are] arguing against one of the more prominent theories of 5-HT 1 action: the behavioral inhibition hypothesis". My concern is that I am not sure whether the findings presented in this paper are extremely novel or resolve a long-standing question, particularly light of many recent studies in humans and monkeys that have long progressed beyond the early ideas of inhibition by Soubrie. From this work, it has become clear that the definition of 'inhibition' as effectively the opposite of behavioural activation is much too limited, and in this light, the current results cannot refute an 'inhibition' explanation of the role of serotonin. Below I will highlight some of this work that the authors should consider when interpreting these results:

- Work on reversal learning by Hannah Clark, showing that frontal serotonin depletion leads to increased perseverative behavior and failures in reversal learning, which are interesting to discuss in relation to the current findings of increased perseverance with increased serotonergic firing

- Perhaps even more directly linked work on alterations in inappropriate 'lose-shift' behavior in probabilistic reversal learning tasks. In these tasks, due to their probabilistic nature, on 20-30% of trials participants need to ignore losses and inhibit shifting to the alternative stimulus, very similar to the current task where the rats need to inhibit shifting to the other

ROI following unrewarded nose pokes when the reward rate is still high. Such alterations of lose-shift behavior have been demonstrated to be associated with changes in serotonin in humans, using behavioural genetics, depressed patients, and direct serotonin manipulations (Chamberlain et al; Science, 2006, Murphy et al, psychological medicine, 2003; den Ouden et al; Neuron, 2013). This idea of serotonin helping to inhibit motivationally driven prepotent responding is also seen in other circumstances e.g. inhibition to respond very quickly when under time pressure (e.g. den Ouden et al. psychopharmacology 2016). Thus, the current findings may in fact refine, rather than refute, theories of serotonin and (motivated) inhibition. Inhibition is likely more than just the absence of motor activation, but rather than active suppression of a prepotent, impulsive response (e.g. leaving the patch). The current findings are still in line with such an inhibition interpretation.

- A recent body of theoretical and empirical work further elaborates on these ideas, that the effect of serotonin should not be understood in terms of just enhancing passivity, but rather is important in linking processing of aversive stimuli to behavioural inhibition (see work by e.g. Huys, Daw, Cools). There are a number of studies that find varying degrees of evidence for this idea (e.g. Geurts et al. Journal of Neuroscience 2013, Crockett et al. 2009, 2012), which should also be discussed

While the current optogenetic findings are certainly interesting in the sense that they afford much higher precision in terms of being able to conclusively claim that the manipulation truly affects serotonin neurons, they do lack in precision in terms of where the serotonin is released. Recent theories and reviews have emphasized the importance of this in reconciling very different effects of serotonin depending on their (amygdala, striatal, frontal) targets (e.g. Deakin 2013 J Psychopharm) and receptors.

MINOR COMMENTS

I fully agree with the authors on the benefits of using formal models of behaviour to infer mechanistic processes and uncover latent, unobservable decision variables. However, I think that the modelling approach in this paper could be improved. Even though the behavioural effects are in line with predictions of MVT, the authors decide that this is a less plausible model than the hazard function model. A more principled approach is to do a formal model comparison of the MVT and the hazard models to show that the latter is indeed a better and more parsimonious explanation of the data. Furthermore, the hazard model is rather descriptive and it is unclear what underlying mechanism serotonin might now be affecting, and indeed the authors refer mostly to 'a latent decision variable' without being more precise.

I find it hard to follow the relevance of the ODC measure. Describe in the methods section how you computed the ODC and how this relates to the hazard function. These two are based on the same behavioural data, so they are not independent measures – then why is their correlation additionally informative?

Why did you use such steep exponential decay functions for the 3 conditions? As a result, after a relatively low number of nose pokes the reward probabilities have dropped to nearly

the same levels for all of the three conditions, which makes the fact that they exit at not significantly different reward rates less surprising. f

For null effects it is important, particularly given the small n , to report Bayesian statistics / evidence in favour of the null hypothesis (e.g. null effect of delta pokes/trial between different condition). The JASP software package is recommended for this.

Figure 5e: does plot e mean that the decreased hazard of leaving is particularly lower for the early trials in the run? How does that explain persistence effects which one would expect to occur later in the trial?

Panel g appears to be missing from figure 6 (but is in the caption)

Make clearer what control analysis the 'shuffled dataset' illustrate, particularly also explain more clearly why there is a significant effect in figure 2h (rather than letting the reader infer)

repeated typo: photostimualtion

Dear Editor and Reviewers,

We are very grateful for the editor and reviewer's constructive comments on the manuscript entitled "Activation of serotonin neurons promotes active persistence in a probabilistic foraging task". We believe we have been able to address all the concerns raised by the reviewers and that this has improved our manuscript. Specific point-by-point replies are as follows:

Reviewer #1 (Remarks to the Author):

Lottem et al. manipulated 5-HT neurons in mice performing a foraging task. This is a nice manuscript, well motivated theoretically, and very timely.

Major comments

1. My main concern is the nagging question of whether the experiment is doing what the authors claim it is. Specifically, what are the effects of 5-HT neuron optogenetic stimulation? Are 5-HT neurons able to follow 25 Hz stimulation for the applied durations? It seems likely that many of the neurons could follow 25 Hz stimulation briefly, followed by a prolonged inhibition (presumably due to autoreceptors). Thus, the observed effects could actually be due to pauses of 5-HT neuron firing (or, possibly, biphasic responses of excitation followed by inhibition). This could be resolved by recording from 5-HT neurons while stimulating them, ideally in the awake mouse, but possibly in the anesthetized mouse.

We agree with the reviewer that this is a potentially important issue. The reviewer's main concern is not failure of stimulation but that we are actually inhibiting rather than exciting 5-HT neurons. While we agree that this is a theoretical possibility, we are not aware of any evidence in the literature that would suggest such a scenario. In fact, in two published papers (Fonseca et al., 2015, Lottem et al., 2016) we showed that photostimulation of DRN 5-HT neurons at increasing pulse frequencies produced a monotonically increasing dose-dependent effect.

In Fonseca et al. we showed that photostimulation resulted in an increase in mice's waiting time in a task that requires them to wait several seconds for a randomly delayed reward. This effect was strongest at 25 Hz photostimulation frequency - the same frequency we used in this paper. And, in Lottem et al. we found that photostimulation of DRN 5-HT neurons led to strong inhibition of spontaneous firing of piriform cortex neurons. This inhibition was monotonically increasing for up to 30 Hz photostimulation frequency, and could be sustained for at least 7 seconds.

These data are hard to reconcile with a pure inhibition of biphasic excitation/inhibition response, since the lower frequencies also used in these studies are very unlikely to

produce inhibition and yet they produced the same sign of functional effects as the 25 Hz trains.

As a more subtle point, high frequency ChR2 stimulation may not result in inhibition but might simply cause unreliable spiking (i.e. less than one spike per stimulation pulse) and that reliability is likely to decrease as a function of stimulation frequency. This would be consistent with a monotonically increasing output function—it would however saturate more quickly than it would have otherwise and might not be effective throughout the stimulation period. While we do not believe that our conclusions would be compromised by such a scenario, to address this issue directly, we performed a new set of experiments in which we recorded DRN electrophysiological responses in SERT-Cre mice expressing ChR2 in the DRN while we applied 25 Hz photostimulation similar to that used in this study. We used a long pulse duration of 15 s, which covers the 95 percentile of trials durations in our experiments. Due to the high frequency and short pulse durations we could not reliably isolate single units, but we obtained high quality multiunit data ($n=7$ multiunits in 2 mice). We found that 5-HT neurons reliably responded to this stimulation protocol throughout the duration of the train. These data are now included in the paper as Supplementary Fig. 5 and referred to on p. 11 of the main text. These data confirm that we do not produce frank inhibition of DRN 5-HT neurons even at 25 Hz and suggest that the degree of sublinearity of the output is even less than might be expected.

Finally, to test whether we might detect a waning of stimulation efficacy over time in the behaviour itself, we performed a new analysis on our data set. We reasoned that if the effect of stimulation on DRN 5-HT spiking turns biphasic or simply rapidly adapts, then we would expect to see less effect toward the end of longer trials, since stimulation always starts at the beginning of a trial and continues until the end. A key variable in this analysis is the number of non-rewarded pokes that a mouse makes after the last rewarded one. This number is increased by DRN 5-HT stimulation. However, we found no correlation between the pokes after last reward and the duration of stimulation. These data are shown in Suppl. Fig. 6 and referred to on p. 11 of the text.

We believe, and hope the reviewer concurs, that together these new data, analysis and previous studies strongly suggest that stimulation effects are sufficiently reliable.

2. How do the present results relate to those recently published from the Mainen lab (Correia et al., eLife, 2017)? Specifically, I wonder whether there are long-term effects of stimulation that can be dissociated from the short-term ones. For example, are pokes per trial different from control when there is a long sequence of consecutive trials with stimulation, immediately following such a sequence? A long time after such a sequence?

We thank the reviewer for suggesting this analysis. We performed the suggested analysis, shown now in Figure 3f, and found that the effect of stimulation did not change as a function of the number of consecutive stimulation trials.

However, it is possible that longer-term effects (on the order of days or even weeks) are taking place, similar to our previously published results. This possibility is of great interest to our lab and we are currently performing experiments to tackle this issue directly, namely performing long term 5-HT stimulations during both foraging and open-field exploration. However, due to their difficulty and because they enter into a new topic, these experiments are beyond the scope of the present study.

3. Please include more details of the stimulation (e.g., a histogram of stimulation durations).

This histogram is now shown in Supplementary figure 5.

4. Are there systematic differences between mice as a function of the amount or extent of ChR2 expression? It would be nice to see more detailed analysis of individual mouse behavior. Related to this, Fig. 1 would benefit from an entire example session (e.g., in the format of 1b or 1d).

We thank the reviewer for suggesting these analyses aimed at describing stimulation's effect on an individual mouse level. We now added the following:

- We performed a more detailed histological analysis, shown in Supplementary Fig. 2. Indeed, we found a significant correlation between the extent of 5-HT expression and the effect of stimulation on behavior.
- We have added two Supplementary Figs. 3 and 8 that show individual mouse data for the two main effects of photostimulation, i.e. increased numbers of pokes and increased vigor.
- Figure 1d now shows an entire example session.

Minor comments

1. I suggest replacing bar plots with histograms or box plots (or something that better represents the distributions). Also, Fig. 2b doesn't appear to have error bars.

We appreciate the reviewer's suggestion. While this concern may be partially a matter of style, we agree that it is important to describe the variability in the data. For this reason, all the plots that demonstrate a significant effect of photostimulation on behavior (3, 5 and 6) also depict individual mouse data points. Additionally, we have added Supplementary Fig. 3, which shows both individual mouse data and the distribution of the effect across the population of mice. Finally, error bars have been added to Figure 2b.

2. Are there enough trials to examine the effects of stimulation on errors?

Unfortunately, we only stimulated during correct trials, so we cannot test for stimulation's effect in error trials.

3. The data may not warrant a title that includes "exploitation," given the subtleties of exploration-exploitation balance, and the extent to which that was tested here.

We agree with the reviewer's suggestion and changed the title to read: "Activation of serotonin neurons promotes active **persistence** in a probabilistic foraging task"

4. Fig. 6 would be clearer if panels (c) and (d) had consistent coloring.

We thank the reviewer for the suggestion and changed the figure accordingly.

5. The model is confusing as written in Fig. 5a. Do the authors mean "logit(h_0)" instead of " h_0 "? Why use "h" instead of " λ " here?

The reviewer is correct. Please note that in the new version of the figures we omitted the equation from figure because it already appears in the text.

6. It would be useful to see, in addition to the subjective reward probability in Fig. 2g, the calculation based on Eq. 1. Similarly, in Fig. 2d, it would be useful to see actual reward rate.

We thank the reviewer for suggesting these analyses. As can be seen in Reviewer figure 1, actual reward probabilities at leaving are different in the three trial types. However, this is a subtle, yet trivial consequence of Equation 2: since subjective probabilities represent a weighted average of three probabilities (corresponding to the three trial types), at any given moment they are higher than the lowest possible (worse trial) and lower than the highest possible probability.

We believe that adding this figure will not add new information to the reader (beyond what is already presented in Figure 2g,h and Equation 2), while it will make the figure harder to understand. We therefore prefer to leave this analysis out of the manuscript.

Reviewer #2 (Remarks to the Author):

Summary

The authors trained 16 mice to perform a novel probabilistic foraging task. 10 of mice expressed Cre under the SERT promoter and other 6 were wild-type litter-mates. All animals had a cre-dependent channel rhodopsin injected into the dorsal raphe nucleus. The task required mice to run back and forth along a linear maze to ports on either end. Pokes into a port resulted in a probabilistic water reward, with the probability decaying exponentially with each poke. Amazingly, mice are near optimal at this task: they leave each port (on average) when the instantaneous probability of reward is equal to their

average reward rate. Stimulation of the 5-HT neurons in the dorsal raphe increase the # of pokes the subjects make before leaving. The behavior, and the stimulation effects, seemed well-fit by a proportional hazards model. The main claim is that the work refutes the hypothesis that 5-HT activation leads to behavioral inhibition.

Overall, I found the paper clearly written and interesting. Taken at face value, the paper would result in a significant shift in the putative role of 5-HT: from "behavioral inhibition" to something more like a "persistence" signal (but maybe more mechanistic, by increasing the time-constants of accumulation to bound processes in general). Of course, a single experiment rarely can (or should) result in a major shift in the field, but I think that others will be inspired to further test this alternative view.

The Mainen lab has been working for some time on neural mechanisms related to "decisions to wait", and I think this paper is a nice step that I imagine (and hope) will lead to linking M2 recording with 5-HT stimulation. The authors posted this paper on Biorxiv which allowed me to discuss the paper with colleagues before the submission of the review. I recommend the paper for publication after my concerns are addressed.

Sincerely,
Jeffrey Erlich

Major Comments

1. Histology (and comparing with variation across animals)

The authors describe their histology in the methods, but did not show the results. It seems two factors observable in the histology might have explained the variation seen in the effects of stimulation across animals. (a) Overall levels of expression, (B) location of the fiber relative to the expression. Are either of these the case?

We thank the reviewer for suggesting this analysis. We performed a more detailed histological analysis, shown in Supplementary figure 2 and found that ChR2 expression, but not fiber location, correlated with photostimulation's effect.

2. Alternative models

The authors acknowledge in their discussion that the cox model is hard to interpret. I think they do an excellent job in the discussion exploring the possibilities. However, it seems (maybe naively) that further modeling could rule out some potential confounds. For example, the authors claim that 5-HT neuron stimulation is not rewarding and doesn't generate conditioned place-preference. However that is directly contradicted by Liu et al 2014 (<http://dx.doi.org/10.1016/j.neuron.2014.02.010>). I was surprised that the authors failed to cite that paper. If 5-HT neurons (via 5-HT and glutamate) were directly rewarding what would the authors expect the result to be in the probabilistic foraging

task? I imagine if it was strongly rewarding then the effect would be much bigger (e.g. animals would just hang out and enjoy the stimulation), but maybe the 5-HT neurons decrease their activity over the long stimulation (something we could see with simultaneous electrophysiology). Can this interpretation be ruled out? Without knowing the effect of the long stimulation duration on the 5-HT neurons it seems that it is quite difficult. One further experiment that would (i think) be very clear is to randomly stimulate on certain pokes rather than during the entire trial. If the stimulation is directly rewarding, it would act like a reset on the run (e.g. like water delivery). But if it just slowed the accumulation of decision-to-leave variable (as the authors claim) then it would have a smaller effect than the whole trial stimulation and no specific effect on poke-microarchitecture.

We thank the reviewer for this comment. We have now elaborated our discussion regarding the possibility of a rewarding effect of DRN 5-HT stimulation (and the reasons for us believing this is not the case, at least in our hands): " If 5-HT modulated the costs or benefits of on-going events with which it was correlated, one might expect this to be revealed in a broad range of situations. Yet, 5-HT stimulation does not produce place preference or bias thigmotaxis in the open field and 5-HT stimulation applied during the outcome of a two-alternative choice value-based decision-task, does not bias choices. While these results are at odds with those reported by Liu et al., which had found a reinforcing effect of optogenetic DRN 5-HT neuron stimulation on behavior, differences in the transgenic mouse lines used to drive ChR2 expression (ePet1-Cre in Liu et al. vs. SERT-Cre here) or the targeting of different anatomical parts of the DRN may account for this apparent discrepancy."

To address the concern of whether stimulation efficacy wanes with time, we performed new experiments in which we photostimulated DRN 5-HT units while performing extracellular recordings in anesthetized mice (see also reply to Reviewer 1, point 1). We found that at 25 Hz, stimulation was effective throughout the stimulation period.

Finally, we performed a new analysis to attempt to detect the effect of a waning stimulation efficacy (see also reply to Reviewer 1, point 1). Based on our modelling of the behavior, specifically the resetting property of the proportional hazards model (Figure 5), the number of omissions mice experience after the last reward within a trial is approximately independent of the trial's history before the last reward. We therefore analyzed the effect of stimulation duration on this quantity (which, as expected, increased during photostimulated trials). Importantly, since the timing of the last reward was variable, ranging from 0 to 20 sec, we could test for a correlation between the duration of photostimulation before the last reward and the number of omissions after the last reward. As shown in Suppl. Fig. 6, we did not observe such a correlation, suggesting that the effect of stimulation on behavior was independent of its duration.

Minor Comments:

1. Reliability of the effects

It seems that the stimulation worked in 7/10 animals. Bar graphs that combine all data across animals do not give readers a good sense of the reliability of behaviors or effects. I would like to see most figures generated for each animal (can be supplemental figures). I would also recommend to the authors to strengthen the sensitivity of their statistics by using mixed-effects models (eg. for fig 2f) instead of ANOVA.

We appreciate the reviewer's suggestion. We agree that in addition to mean effects, it is also important to show the variability in the data. For this reason, all the plots that show a significant effect of photostimulation on behavior (3, 5 and 6) now depict individual mouse data points that allow the reader to gauge the magnitude and variability of the effects. Additionally, we have now added Supplementary Fig. 3, which shows both individual mouse data and the distribution of photostimulation's effect across the population of mice. Finally, error bars have been added to Fig. 2b.

We also agree with the reviewer that mixed-effects models are generally superior to ANOVA, the differences that make them so do not apply to the analysis performed in Fig. 2. Namely, the predictor (trial type) is not continuous, we did not have missing data points, and we had no relevant groupings of the mice. We therefore believe that the simple ANOVA is preferred in this situation.

2. Model comparison.

The authors did not perform standard model comparison techniques, like leaving out parameters and checking AIC, BIC, DIC or MDL (minimum descriptive length). In others words, they showed one model that seemed to fit well, but didn't try any other models or show which parameters were "necessary".

We thank the reviewer for suggesting this analysis. We have now added formal model comparison in new Supplementary Fig. 4. In this analysis, we first compare the Cox regression model with a simpler one that follows MVT's prediction. The MVT model assumes that leaving decision are made stochastically based on the estimated reward probability. Additionally, we compare different versions of the Cox model having different numbers of parameters. Using AIC as our criterion, we found that the full Cox mode outperforms all the above alternatives.

3. on line 385, in the discussion, it would be nice for the authors to help a less experienced reader out by mentioning some of the other major roles of 5-HT with some cited reviews: e.g. depression, sleep, digestion.

We thank the reviewer for this suggestion and have added the following line to the discussion: "... then the range of behaviors in which 5-HT acts, such as waiting, cognitive flexibility and sensorimotor gain control,..."

4. Has anyone previously shown that mice are capable of near optimal foraging? The authors might want to emphasize that aspect of the results a bit more.

We thank the reviewer for this suggestion and have added the following to the introduction on p.4: "There are a number of studies of optimal foraging in rodents in their natural habitats, and it has been proposed that certain operant behaviors are closely related to foraging behavior, but the use of foraging behaviors in a laboratory setting remains a relatively underexplored area (but see Refs. 21,22)."

5. PFT is not a standard acronym, i would just find/replace with probabilistic foraging task. or in many cases you can just say "the task". Sometimes you say "the PFT" other times you omit the "the".

We have omitted the acronym PFT from the paper.

6. Difference between 5-HT neurons and 5-HT. The authors are stimulating 5-HT neurons but those neurons express glutamate as well. They authors should be more careful in their use of 5-HT vs. 5-HT neurons in the DRN.

We agree with the reviewer that care should be taken when extrapolating our finding from effects of DRN 5-HT stimulation to the endogenous function of 5-HT, particularly since these neurons are known to co-release other neurotransmitters, such as glutamate. We have changed the text accordingly.

7. Was there an increase in errors (return to the just visited poke) after stimulation? If so, that would support a reinforcing effect of the stimulation.

As can be seen in Reviewer figure 2, stimulation on given trial did not affect performance (i.e. the probability of correctly alternating to the other side) on the subsequent trial.

8. Was there any relationship between the individual variation in baseline behavior and effect of stimulation? It would be nice to see figure 2d, but with a second blue dot for the 10 SERT animals under stimulation. According to the "main effect" the blue dots should shift down relative to the black dots. Is this the case? I guess it might be very noisy.

We thank the reviewer for suggesting this analysis. We tested for a correlation between baseline behavior and the effect of stimulation and found none, as can be seen in Supplementary figure 3. As for the suggested figure, we note that since the reward probability is monotonically decreasing with the number of pokes, any manipulation that increases the number of pokes (such as DRN 5-HT stimulation) would necessarily result in lowering the reward probability at leaving. As the reviewer notes, adding these data to the existing figure is noisy (Reviewer figure 3), and for these reasons decided to not add it to the manuscript.

Reviewer #3 (Remarks to the Author):

This study addresses the long-standing question of the role of serotonin in motivated decision-making. Specifically, the authors use a foraging task in combination with optogenetic activation of 5HT neurons to assess the willingness of mice to explore a depleting reward site, where they aim to pit against each other hypotheses of serotonin's role being characterised as behavioural inhibition (putatively indexed by leaving a patch early) versus patience / persistence. They show that the mice's persist longer in a depleting patch when their serotonin neurons are stimulated. I believe that this is an interesting finding and the role of serotonin in decision-making is indeed an important question. Optogenetic approaches allow for a new level of neurochemical specificity which particularly serotonin research has been lacking.

The authors' main claim is that (line 340) "our results [are] arguing against one of the more prominent theories of 5-HT 1 action: the behavioral inhibition hypothesis". My concern is that I am not sure whether the findings presented in this paper are extremely novel or resolve a long-standing question, particularly light of many recent studies in humans and monkeys that have long progressed beyond the early ideas of inhibition by Soubrie. From this work, it has become clear that the definition of 'inhibition' as effectively the opposite of behavioural activation is much too limited, and in this light, the current results cannot refute an 'inhibition' explanation of the role of serotonin. Below I will highlight some of this work that the authors should consider when interpreting these results:

1. Work on reversal learning by Hannah Clark, showing that frontal serotonin depletion leads to increased perseverative behavior and failures in reversal learning, which are interesting to discuss in relation to the current findings of increased perseverance with increased serotonergic firing. Perhaps even more directly linked work on alterations in inappropriate 'lose-shift' behavior in probabilistic reversal learning tasks. In these tasks, due to their probabilistic nature, on 20-30% of trials participants need to ignore losses and inhibit shifting to the alternative stimulus, very similar to the current task where the rats need to inhibit shifting to the other ROI following unrewarded nose pokes when the reward rate is still high. Such alterations of lose-shift behavior have been demonstrated to be associated with changes in serotonin in humans, using behavioural genetics, depressed patients, and direct serotonin manipulations (Chamberlain et al; Science, 2006, Murphy et al, psychological medicine, 2003; den Ouden et al; Neuron, 2013). This idea of serotonin helping to inhibit motivationally driven prepotent responding is also seen in other circumstances e.g. inhibition to respond very quickly when under time pressure (e.g. den Ouden et al. psychopharmacology 2016). Thus, the current findings may in fact refine, rather than refute, theories of

serotonin and (motivated) inhibition. Inhibition is likely more than just the absence of motor activation, but rather than active suppression of a prepotent, impulsive response (e.g. leaving the patch). The current findings are still in line with such an inhibition interpretation.

We appreciate the reviewer's thorough suggestions regarding our discussion of, and conclusions from, the results of our study. Indeed, in addition to its similarities to waiting tasks, probabilistic foraging also bares resemblance to probabilistic reversals tasks, in which it was shown that lower 5-HT levels correspond to increased lose-shift behavior. We now added the following text to the discussion on p.20:

" ... This interpretation can explain the observation that DRN photostimulation does not bias travel times between reward sites when stimulation occurs after the mouse is already in transit (Fig. 4 and ref. 47). It would also be consistent with an observed increase in active escape behavior (reduction of immobility) in the forced swim test that is induced by stimulation of medial prefrontal cortex axons in the DRN and enhanced lose-shift behavior in probabilistic reversal tasks, observed in both patients with major depression and in individuals homozygous for the long serotonin transporter allele, both associated with decreased levels of extrasynaptic serotonin.."

2. A recent body of theoretical and empirical work further elaborates on these ideas, that the effect of serotonin should not be understood in terms of just to enhancing passivity, but rather is important in linking processing of aversive stimuli to behavioural inhibition (see work by e.g. Huys, Daw, Cools). There are a number of studies that find varying degrees of evidence for this idea (e.g. Geurts et al. Journal of Neuroscience 2013, Crockett et al. 2009, 2012), which should also be discussed.

We have now elaborated our discussion on this issue as follows: "In contrast, elevated 5-HT levels have also been associated with aversive processing. For example, acute tryptophan depletion (an experimental manipulation used to lower 5-HT levels in humans), attenuates punishment-induced inhibition, suggesting that endogenous 5-HT release may reduce response vigor in the face of aversive predictions. However, these effects were shown to be specific to aversive contexts, which were absent in the current study. Such context dependent differences in 5-HT's effect on foraging behavior could be tested in future experiments by, for example, introducing probabilistic punishments during resource exploitation, and testing the effect of DRN 5-HT stimulation on response vigor under these conditions."

3. While the current optogenetic findings are certainly interesting in the sense that they afford much higher precision in terms of being able to conclusively claim that the manipulation truly affects serotonin neurons, they do lack in precision in terms of where the serotonin is released. Recent theories and reviews have emphasized the importance of this in reconciling very different effects of serotonin depending on their (amygdala, striatal, frontal) targets (e.g. Deakin 2013 J Psychopharm) and receptors.

We appreciate the reviewers comment, and acknowledge that teasing apart the various pathways through which 5-HT exerts its modulatory effects is one of the major challenges facing the field of 5-HT research (a challenge we cannot address with our dataset). Indeed, we are currently conducting experiments aimed exactly at this issue, targeting different DRN 5-HT projections and stimulating them directly during foraging behavior. However, the results from these experiments are preliminary, and go beyond the scope of the current manuscript.

MINOR COMMENTS

1. I fully agree with the authors on the benefits of using formal models of behaviour to infer mechanistic processes and uncover latent, unobservable decision variables. However, I think that the modelling approach in this paper could be improved. Even though the behavioural effects are in line with predictions of MVT, the authors decide that this is a less plausible model than the hazard function model. A more principled approach is to do a formal model comparison of the MVT and the hazard models to show that the latter is indeed a better and more parsimonious explanation of the data. Furthermore, the hazard model is rather descriptive and it is unclear what underlying mechanism serotonin might now be affecting, and indeed the authors refer mostly to 'a latent decision variable' without being more precise.

We thank the reviewer for suggesting this analysis. We have now added formal model comparison in Supplementary Fig. 4. In this analysis, we first compared the Cox regression model with a model that follows the predictions of the marginal value theorem. This model assumes that leaving decision are made stochastically based on the estimated reward probability. Additionally, we compared different versions of the Cox model having different numbers of parameters. Using the Akaike Information Criterion (AIC), we show that the full Cox model outperforms all the above alternatives.

2. I find it hard to follow the relevance of the ODC measure. Describe in the methods section how you computed the ODC and this relates to the hazard function. These two are based on the same behavioural data, so they are not independent measures – then why is their correlation additionally informative?

The ODC was computed by dividing each nose-poke's duration by the sum of the duration and the preceding inter-poke-interval, this was now added to the main text, line 244. Importantly, the hazard was calculated solely based on poke counts (discarding all temporal information), whereas the ODC was calculated based on actual, real-time measurements, but ignoring poke counts. Furthermore, while normative/optimal framing of the decision (e.g. MVT) predicted that the hazard of leaving would decrease as a function of reward probability, we did not have similar predictions regarding the micro-structure of nose-poking behavior (ODC) – making the two measures independent, at

least a-priori. Therefore, we believe the finding that the two were correlated was both novel and non-trivial.

3. Why did you use such steep exponential decay functions for the 3 conditions? As a result, after a relatively low number of nosepokes the reward probabilities have dropped to nearly the same levels for all of the three conditions, which makes the fact that they exit at not significantly different reward rates less surprising.

This concern prompted us to perform the shuffling analysis in Figure 2f,h. If the three exponential functions were decaying so rapidly such that they would be indistinguishable at the time of leaving then shuffling the data with respect to trial type would not affect these results – that is, probabilities at leaving would be equal (irrespective of trial type), and the real and shuffled data would be indistinguishable. However, we did find significant differences in the reward probability at the time of leaving between the different trial types in the shuffled data. If mice were using a timing-base strategy or some other reward-independent assessment of leaving time, reward rates would not be similar. This suggests that the mice adaptively changed their behavior (by staying longer in better trials) such that the probabilities at leaving were similar for the different conditions.

4. For null effects it is important, particularly given the small n, to report Bayesian statistics / evidence in favour of the null hypothesis (e.g. null effect of delta pokes/trial between different condition). The JASP software package is recommended for this.

We thank the reviewer for this comment. To address this concern, we have added Supplementary Fig. 3 that shows individual mouse data and distributions of the main-effect's values and significance. We found that in addition to the significance of the observed effect, as calculated using standard non-parametric methods, when analyzed individually, 14 out of 16 SERT mice showed significant increased persistence during stimulated trials, in contrast to 0 out of 6 control mice.

5. Figure 5e: does plot e mean that the decreased hazard of leaving is particularly lower for the early trials in the run? How does that explain persistence effects which one would expect to occur later in the trial?

The effects of regressors in the cox proportional hazards model is, by definition, multiplicative. As a result, values that are further away from zero seem to be more affected in absolute terms. Therefore, to the extent that this model accurately represents an underlying decision process, our interpretation is that stimulation's effect is constant (and divisive) throughout the trial, and that the interaction between this constant effect and a dynamically changing hazard ultimately gives rise to the observed effect of stimulation. Additionally, as can be seen in Supplementary figure 6, the effect of stimulation on the number of nose-pokes is independent of trial length.

6. Panel g appears to be missing from figure 6 (but is in the caption)

That is correct. We thank the reviewer and we have corrected the caption accordingly.

7. Make clearer what control analysis the 'shuffled dataset' illustrate, particularly also explain more clearly why there is a significant effect in figure 2h (rather than letting the reader infer)

We thank the reviewer for this suggestion and have added the following explanation to the main text on p.7: "If mice were using a reward-independent assessment of leaving time, reward probabilities at the time of leaving would not be similar across the different trial types. Instead, the comparison between shuffled and real data confirms that the mice were sensitive to the statistics of individual port visits and suggests that the mice employed a near-optimal strategy in this task."

8. repeated typo: photostimualtion

We thank the reviewer for catching this. The text was corrected accordingly.

Reviewer figures

Reviewer figure 1

- (a) Cumulative distributions of the actual reward probability after the last poke in a trial (i.e. at the time of switching) for the three trial types, averaged across mice.
- (b) Cumulative distributions of the estimated subjective reward probability (following Equation 2) after the last poke in a trial for the three trial types, averaged across mice.
- (c) Bar plot showing the average actual and estimated reward probabilities after the last poke.

Reviewer figure 2

Bar plot showing the average performance after stimulated and non-stimulated trials.

Reviewer figure 3

Scatter plot of reward rate at leaving in stimulated (blue) and control (gray) trials vs. average reward rate. Each circle represents one SERT-Cre mouse.

References

Lottem E, Lörincz ML, Mainen ZF. Optogenetic activation of dorsal raphe serotonin neurons rapidly inhibits spontaneous but not odor-evoked activity in olfactory cortex. *J Neurosci*. 2016.

Fonseca MS, Murakami M, Mainen ZF. Activation of dorsal raphe serotonergic neurons promotes waiting but is not reinforcing. *Curr Biol*. 2015.

REVIEWERS' COMMENTS:

Reviewer #1 (Remarks to the Author):

The authors have addressed my concerns. This is an excellent paper.

Reviewer #2 (Remarks to the Author):

Lottem et al, Resub

The authors responded adequately to the reviewers concerns.
I recommend the paper for submission after addressing the following minor revisions.

Comments

Supp Fig 3: Color choice is pretty bad. very hard to see blue vs. black. I think the x-axis in panel b has an error (-1.25 0 -1.25).

Supp Fig 6: are the differences in b & c significant? The captions could be more informative.

Reviewer #2 comments

Supp Fig 3: Color choice is pretty bad. very hard to see blue vs. black. I think the x-axis in panel b has an error (-1.25 0 -1.25).

We thank the reviewer and have changed the colors and increased the line widths to make Supp Fig 3 clearer. We also corrected the x-axis label accordingly.

Supp Fig 6: are the differences in b & c significant? The captions could be more informative.

The differences were indeed significant and this information was added to the caption.